# Intravital quantification reveals dynamic calcium concentration changes across B cell differentiation stages

Carolin Ulbricht[1,2], Ruth Leben[3], Asylkhan Rakhymzhan[3], Frank Kirchhoff[4], Lars Nitschke[5], Helena Radbruch[6], Raluca A Niesner[3,7†], Anja E Hauser[1,2†]*

[1]Charité – Universitätsmedizin Berlin, corporate member of Freie Universität Berlin and Humboldt-Universität zu Berlin, Department of Rheumatology and Clinical Immunology, Charitéplatz 1, Berlin, Germany; [2]Immune Dynamics, Deutsches Rheuma-Forschungszentrum Berlin, ein Institut der Leibniz-Gemeinschaft, Berlin, Germany; [3]Biophysical Analytics, Deutsches Rheuma-Forschungszentrum, ein Institut der Leibniz-Gemeinschaft, Berlin, Germany; [4]Universität des Saarlandes, Saarbrücken, Germany; [5]Friedrich-Alexander-Universität Erlangen-Nürnberg, Erlangen, Germany; [6]Charité – Universitätsmedizin Berlin, corporate member of Freie Universität Berlin and Humboldt-Universität zu Berlin, Department of Neuropathology, Charitéplatz 1, Berlin, Germany; [7]Veterinary Medicine, Freie Universität Berlin, Berlin, Germany

*For correspondence:
hauser@drfz.de

†These authors contributed equally to this work

**Abstract** Calcium is a universal second messenger present in all eukaryotic cells. The mobilization and storage of $Ca^{2+}$ ions drives a number of signaling-related processes, stress–responses, or metabolic changes, all of which are relevant for the development of immune cells and their adaption to pathogens. Here, we introduce the Förster resonance energy transfer (FRET)-reporter mouse YellowCaB expressing the genetically encoded calcium indicator TN-XXL in B lymphocytes. Calcium-induced conformation change of TN-XXL results in FRET-donor quenching measurable by two-photon fluorescence lifetime imaging. For the first time, using our novel numerical analysis, we extract absolute cytoplasmic calcium concentrations in activated B cells during affinity maturation in vivo. We show that calcium in activated B cells is highly dynamic and that activation introduces a persistent calcium heterogeneity to the lineage. A characterization of absolute calcium concentrations present at any time within the cytosol is therefore of great value for the understanding of long-lived beneficial immune responses and detrimental autoimmunity.

## Introduction

During generation of humoral immunity to pathogens, calcium-mobilizing events in lymphocytes can communicate such diverse outcomes as migration, survival, stress responses, or proliferation. An elevation of cytoplasmic calcium from external space is mostly mediated through ligand binding to surface receptors (*Baba et al., 2014*). Especially in the germinal center (GC), where B cells fine-tune their B cell receptor (BCR) in order to become positively selected by T cells, ligand density in the form of native antigen (AG), stimuli for toll-like receptors (TLR), or chemokine receptors, is high. Selected B cells that leave the GCs fuel the pool of memory B cells and long-lived plasma cells (LLPCs). These cells produce high-affinity antibodies granting up to lifelong protection against threats such as infectious diseases, but also can account for the persistence of an autoimmune phenotype, when selection within the GC is impaired (*Berek et al., 1991*; *da Silva et al., 1998*; *Hiepe et al., 2011*; *Victora and Nussenzweig, 2012*). B cell activation by AG uptake through the

BCR promotes calcium influx into B cells (*Tolar et al., 2009*). Calcium mobilization eventually switches on effector proteins and transcription factors like nuclear factor kappa B, nuclear factor of activated T cells, or myelocytomatosis oncogene cellular homolog, thereby inducing differentiation events and remodeling of metabolic requirements (*Crabtree and Olson, 2002*; *Jellusova, 2018*; *Luo et al., 2018*; *Saijo et al., 2002*; *Su et al., 2002*). Dependent on the amount of AG taken up and the quality of major histocompatibility complex II (MHCII)-mediated presentation to T follicular helper cells, B cells receive additional, costimulatory signals (*Gitlin et al., 2015*). Interestingly, recent studies propose that costimulatory signals have to occur within a limited period of time after initial BCR activation and that the limit is set by a calcium threshold, eventually leading to mitochondrial dysfunction (*Akkaya et al., 2018*). Thus, quantification of changes in absolute cytoplasmic calcium concentration tolerated by GC B cells would help to understand how B cell selection in the GC is accomplished.

In contrast to qualitative description, absolute calcium measurements in B cells have not yet been performed in vivo, partly because of the lack of internal concentration standards. Two-fluorophore genetically encoded calcium indicators (GECI) relying on Förster resonance energy transfer (FRET) can take on a calcium-saturated (quenched) and calcium-unsaturated (unquenched) condition, overcoming this issue. However, intravital application of quantitative FRET has been hampered by light distortion effects in deeper tissue. The differential scattering and photobleaching properties of the two fluorophores would lead to a false bias towards a higher quenching state. We here introduce a single-cell fluorescence lifetime imaging (FLIM) approach for absolute calcium quantification in living organisms that is tissue depth-independent. Both time-domain and frequency-domain FLIM technologies have been employed in the past 30 years to sense changes in *pH*, ionic strength, $pO_2$, metabolism, and many other cellular parameters within living cells. Due to the impact of these parameters on the chemical structure, the fluorescence lifetime of the analyzed fluorophores can change (*Elson et al., 2004*; *Lakowicz et al., 1992*; *Le Marois and Suhling, 2017*). Particularly, the versatility of FLIM to quantify FRET quenching has been demonstrated and applied in various biological contexts (*Chen et al., 2003*; *Levitt et al., 2020*; *Mossakowski et al., 2015*). The enhanced cyan fluorescent protein (eCFP)/citrine-FRET pair-GECI TN-XXL is able to measure fluctuations in cytoplasmic calcium concentration through the calcium binding property of the muscle protein troponin C (TnC) (*Mank et al., 2006*). Calcium binding to the fluorophore-linker TnC quenches eCFP fluorescence through energy transfer to citrine (FRET), linking decreasing eCFP fluorescence lifetime to increasing calcium concentration. Whereas eCFP fluorescence lifetime changes with refractive index (*Strickler and Berg, 1962*) and may change upon large shifts in *pH* value, ionic strength, oxygenation, or temperature, these parameters hardly vary in the cytosol of living cells. Thus, we expect only changes in cytosolic calcium concentration to have an impact on the fluorescence lifetime of eCFP as donor in the TN-XXL construct. In addition, phasor analysis of FLIM data elegantly condenses multicomponent fluorescent decay curves into single vector-based information (the phasor) (*Digman et al., 2008*). For calcium concentration analysis in microscopic images, we first took advantage of the previously published titration curve of TN-XXL by Geiger et al., which we also confirmed in our experimental setup (*Geiger et al., 2012*). We further adapted the phasor-based calibration strategy to quantify calcium levels in vivo proposed by Celli and colleagues to the TN-XXL construct expressed in B lymphocytes (*Celli et al., 2010*). With this method, we are able to describe short- and long-term changes in absolute calcium concentrations within B cells during affinity maturation and differentiation into antibody-producing plasma cells.

We here describe the calcium reporter mouse strain 'YellowCaB' (termed after energy transfer to the *yellow* fluorescent protein citrine in case of *ca*lcium present in the cytosol of *B* cells). These mice express cytosolic TN-XXL in all CD19-positive cells. Intravital FLIM of adoptively transferred Yellow-CaB cells shows that calcium concentrations are highly dynamic in B cells involved in the GC reaction. We describe different patterns of calcium fluctuation regarding amplitude and baseline within non-activated and AG experienced cells and plasma blasts. We observe the emergence of $Ca^{2+}$-high differentiated B cells and plasma blast populations, which might point to cells undergoing metabolic stress.

# Results

## YellowCaB: a system for FRET-based calcium analysis in B cells

Mice expressing a loxP-flanked STOP sequence followed by the TN-XXL-construct inserted into the ROSA26 locus were crossed with the CD19-Cre strain (*Rickert et al., 1997*). The offspring had exclusive expression of the GECI TN-XXL in CD19[+] B lymphocytes, as confirmed by visualization of eCFP and citrine fluorescence by confocal microscopy after magnetic B cell isolation (*Figure 1a, b*). These YellowCaB cells were excited with a 405 nm laser that is capable of exciting eCFP but not citrine. The detection of yellow emission thus can be attributed to baseline FRET representing steady-state calcium levels. Expression of TN-XXL in YellowCaB mice was further confirmed by flow cytometry after excitation with the 488 nm laser and detection in a CD19[+]GFP[+](green fluorescent protein) gate that would also detect citrine fluorescence. Citrine was found to be present in a substantial part of CD19[+] B lymphocytes and was not detectable in the CD19[-] population (*Figure 1c*). *Cd19^{cre/+}* mice heterozygous for TN-XXL and *Cd19^{cre/+}* mice homozygous for TN-XXL did not differ in the

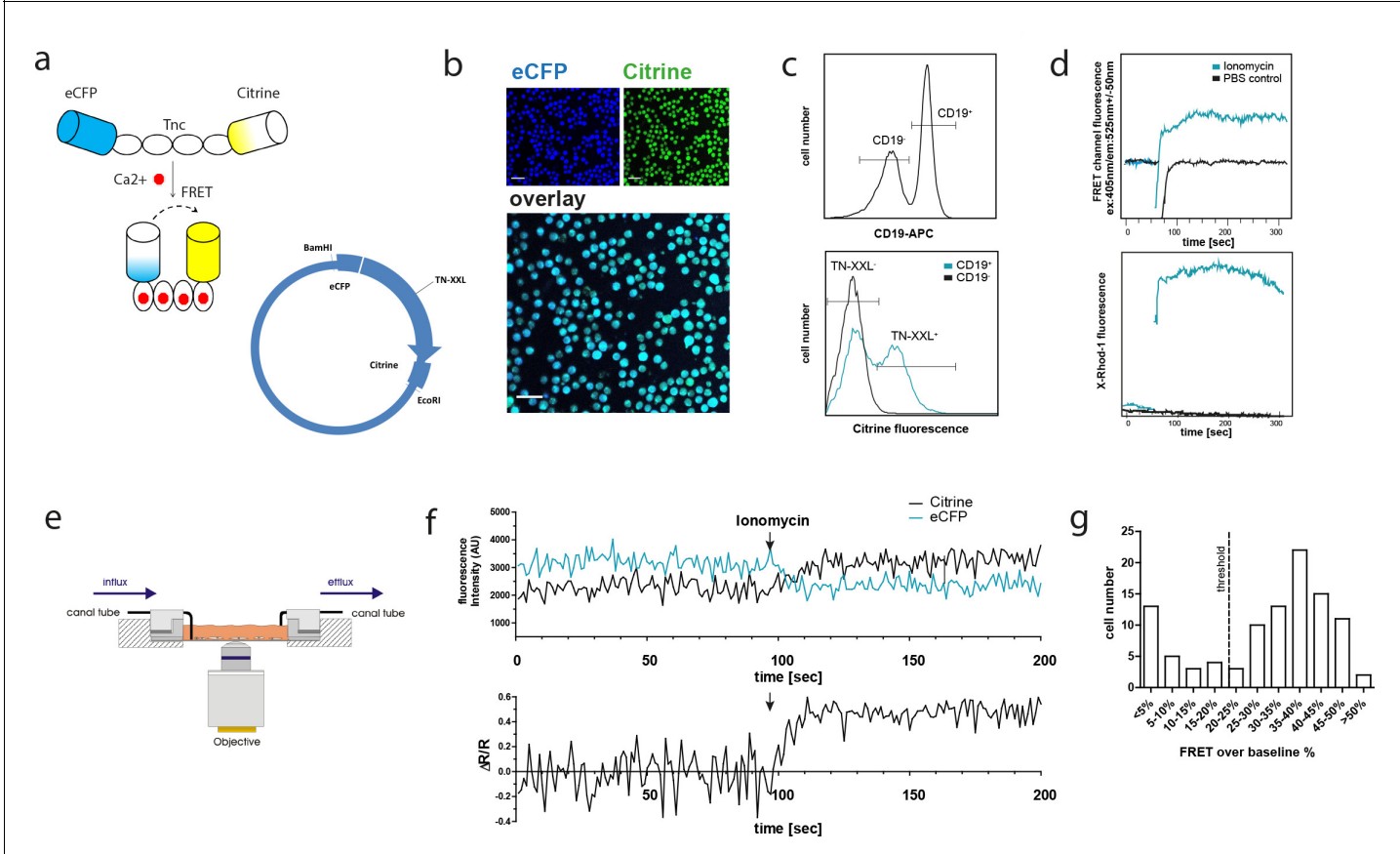

**Figure 1.** The genetically encoded calcium indicator (GECI) TN-XXL is functionally expressed in CD19[+] B cells of YellowCaB mice. (**a**) Schematic representation of the genetically encoded calcium indicator TN-XXL with the calcium-sensitive domain troponin C (TnC) fused to donor fluorophore eCFP and acceptor fluorophore citrine. Binding of Ca$^{2+}$ ions within (up to) four loops of TnC leads to quenching of eCFP and Förster resonance energy transfer (FRET) to citrine. (**b**) Confocal image of freshly isolated CD19[+] B cells. Overlapping blue and yellow-green fluorescence of eCFP and citrine, respectively, can be detected after Cre-loxP-mediated expression of the TN-XXL vector in YellowCaB mice. (**c**) Flow cytometric analysis of TN-XXL expression in lymphocytes from YellowCaB mice. (**d**) Flow cytometric measurement of calcium flux after addition of ionomycin and phosphate buffered saline control. (**e**) Continuous perfusion imaging chamber for live cell imaging. (**f**) Confocal measurement of mean fluorescence intensity and FRET signal change after addition of ionomycin to continuously perfused YellowCaB cells. Data representative for at least 100 cells out of three independent experiments. (**g**) Frequency histogram of > 100 YellowCaB single cells, FRET analyzed after ionomycin stimulation. Threshold chosen for positive FRET signal change = 20% over baseline intensity.

The online version of this article includes the following figure supplement(s) for figure 1:

**Figure supplement 1.** Genotyping of YellowCaB mice and cell numbers.

proportion of cells within the CD19$^+$GFP$^+$ population, nor did male and female mice (*Figure 1—figure supplement 1*). In addition, no differences in total cell numbers and B cell numbers between *Cd19$^{cre/+}$Tn-xxl$^{+/-}$*, *Cd19$^{cre/+}$ Tn-xxl$^{+/+}$*, and wild-type mice were detected (*Figure 1—figure supplement 1*). We next set out to test if we could induce a FRET signal change under calcium-saturating conditions in the cytoplasm. The ionophore ionomycin is commonly used as positive control in in vitro experiments measuring calcium concentrations as it uncouples the increase of calcium concentration from the physiological entry sites of Ca$^{2+}$ ions by forming holes in the cell membrane. When stimulated with ionomycin, a steep increase of the FRET level over baseline was recorded by flow cytometry in the GFP-channel after excitation with the 405 nm laser. Calcium-dependence of the signal increase was further independently confirmed by staining with the calcium sensitive dye X-Rhod-1, that shows a red fluorescence signal increase after calcium binding (*Figure 1d*; *Li et al., 2003*).

In preparation of our intravital imaging experiments, we first tested if the YellowCaB system is stable enough for time-resolved microscopic measurements and sensitive enough for subtle cytoplasmic calcium concentration changes as they occur after store-operated calcium entry (SOCE). In SOCE, stimulation of the BCR with AG leads to drainage of intracellular calcium stores in the endoplasmic reticulum (ER), which triggers calcium influx from the extracellular space into the cytosol through specialized channels (*Baba et al., 2014*). We established a customizable perfusion flow chamber system to monitor and manipulate YellowCaB cells over the duration of minutes to hours (*Figure 1e*). Division of the fluorescence intensity of electron acceptor citrine by that of donor eCFP yields the FRET ratio (R), which is then put into relationship to baseline FRET levels. As expected, we detected a decrease of the eCFP signal, concurrent with an increased citrine fluorescence after the addition of 4 µg/ml ionomycin to continuous flow of 6 mM Krebs–Ringer solution. Overall, this resulted in a maximal elevation of ΔR/R of 50–55% over baseline (*Figure 1f*). Analysis of >100 cells showed that approximately in three quarters of the cells we were able to detect FRET in response to ionomycin treatment, and that the majority of these cells showed 35–40% FRET signal change. According to the two populations visible in the histogram, we defined a change of 20% ΔR/R as a relevant threshold for the positive evaluation of responsiveness (*Figure 1g*). In conclusion, we achieved the functional and well-tolerated expression of TN-XXL exclusively in murine CD19$^+$ B cells for measurement of changes of cytoplasmic calcium concentrations.

## Repeated BCR stimulation results in fluctuating cytoplasmic calcium concentrations

SOCE in B cells can be provoked experimentally by stimulation of the BCR with multivalent AG, for example, anti-Ig heavy chain F(ab)$_2$ fragments. To test the functional performance of the GECI TN-XXL in YellowCaB cells, we stimulated isolated YellowCaB cells with 10 µg/ml anti-IgM F(ab)$_2$ fragments to activate the BCR. In an open culture imaging chamber, we induced an elevated FRET signal with a peak height of >30% that lasted over 3 min (*Figure 2a*). The signal declines after this time span, probably due to BCR internalization or the activity of ion pumps. We tested antibody concentrations at 2, 4, 10, and 20 µg/ml. An antibody concentration of 2 µg/ml was not enough to provoke calcium flux (data not shown), whereas at 4 µg/ml anti-IgM-F(ab)$_2$ we observed 20% an elevated ΔR/R over baseline (*Figure 2b*). At 20 µg/ml anti-IgM-F(ab)$_2$, we detected no further FRET increase (*Figure 2—figure supplement 1a, b*). Thus, we conclude a concentration dependency of the GECI TN-XXL and saturating conditions at 10 µg/ml BCR heavy chain stimulation. Interestingly, the reaction is not completely cut off after the FRET signal has declined, but a residual FRET signal of about 7% compared to baseline values was measured for approximately 3.5 additional minutes (*Figure 2a*). Thus, B cells seem to be able to store extra calcium within the cytoplasm for some time. We therefore wondered if it is possible to stimulate YellowCaB cells more than once. For this purpose, we connected our imaging culture chamber to a peristaltic pump and took advantage of the fact that under continuous perfusion with Ringer solution the flow will dilute the antibody out of the chamber. This way, it is possible to stimulate B cells several times rapidly and subsequently, before BCRs are internalized, indicated by multiple peaks in ΔR/R (*Figure 2b*). In addition, stimulation of the BCR light chain using an anti-kappa antibody led to calcium increase within YellowCaB cells (*Figure 2—figure supplement 1a*). Of note, the resulting FRET peak is shaped differently, and concentrations > 150 µg/ml antibody were needed in order to generate a response.

Since T cell engagement and the binding of microbial targets to innate receptors like TLRs have also been described to raise cytoplasmic calcium in B cells (*Ojaniemi et al., 2003*; *Pone et al.,*

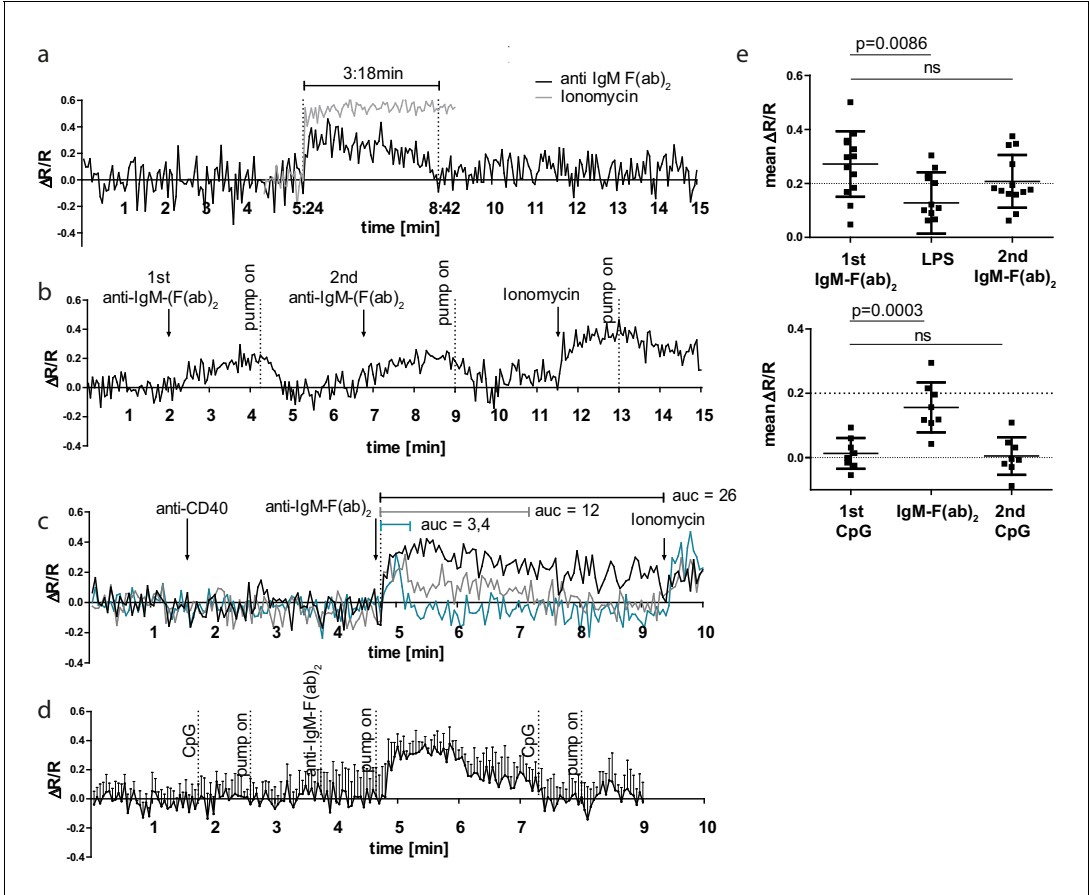

**Figure 2.** B cell receptor (BCR) stimulation specifically leads to calcium mobilization in YellowCaB cells in vitro. (a) Confocal measurement of Förster resonance energy transfer (FRET) duration ($\Delta R/R > 0$) in non-perfused primary polyclonal YellowCaB cells after addition of 10 µg/ml anti-IgM-F(ab)$_2$ (black) and ionomycin control (gray). Data representative for at least 35 single cells in four independent experiments. (b) Confocal measurement of FRET signal change after repeated addition of anti-IgM-F(ab)$_2$ to perfused primary polyclonal YellowCaB cells. Data representative for at least 50 cells out of five independent experiments. (c) Confocal measurement of FRET signal change after addition of anti-IgM-F(ab)$_2$ to perfused primary polyclonal YellowCaB cells following stimulation with anti-CD40 antibody and ionomycin as positive control. Examples of transient cytoplasmic (blue), intermediate (gray), and sustained calcium mobilization shown, area under the curve compared. Data representative for 26 cells out of two independent experiments. (d) Resulting FRET curve out for n = 7 primary polyclonal YellowCaB cells perfused with toll-like receptor (TLR)9 stimulator cytosine phosphate guanine (CpG) in Ringer solution and subsequent addition of anti-IgM-F(ab)$_2$. (e) Mean FRET signal change over time after addition of TLR4 or TLR9 stimulation in combination with BCR crosslinking by anti-IgM-F(ab)$_2$ in perfused polyclonal YellowCaB cells. n = 12 (top) and n = 8 (bottom), one-way ANOVA. Error bars: SD/mean.

The online version of this article includes the following figure supplement(s) for figure 2:

**Figure supplement 1.** Confocal measurement and plot of TN-XXL $\Delta R/R$ over time.

---

2015; Ren et al., 2014), we investigated the response of YellowCaB cells after incubation with anti-CD40 antibodies, as well as the TLR4 and TLR9 stimuli lipopolysaccharide (LPS) and cytosine-phosphate-guanine-rich regions of bacterial DNA (CpG), respectively. Within the same cells, we detected no reaction to anti-CD40 treatment alone, but observed three types of shapes in post-CD40 BCR-stimulated calcium responses, which differed from anti-CD40-untreated cells (*Figure 2a*). These calcium flux patterns were either sustained, transient, or of an intermediate shape (*Figure 2c*). Sustained calcium flux even saturated the sensor at a level comparable to that achieved by ionomycin treatment. Cells that showed only intermediate flux maintained their ability to respond to ionomycin treatment at high FRET levels, as demonstrated by the $\Delta R/R$ reaching 0.4 again after stimulation (*Figure 2c*). Furthermore, integrated TLR and BCR stimulation affected the appearance of the calcium signal. The addition of TLR9 stimulus CpG alone had no effect on YellowCaB FRET levels; however, the subsequent FRET peak in response to anti-Ig-F(ab)$_2$ was delayed (*Figure 2d, e*).

TLR4 stimulation via LPS could elevate calcium concentration of B cells, but only to a minor extent (*Figure 2e*). When TLR4 stimulation by LPS was performed before BCR stimulation, decreased FRET levels in response to anti-IgM-F(ab)$_2$ were observed. We conclude that, in order to become fully activated, B cells are able to collect and integrate multiple BCR-induced calcium signals and that signaling patterns are further shaped by innate signals or T cell help. BCR inhibition abolishes a FRET signal change in response to anti-IgM-F(ab)$_2$ (*Figure 2—figure supplement 1b*). Of note, we excluded the possibility that measured signal changes were related to chemokine stimulation. In vitro, we detected no FRET peak after applying CXCL12, probably because of lacking GECI sensitivity to small cytoplasmic changes (*Figure 2—figure supplement 1c, d*). Thus, the YellowCaB system is well suited for the detection of BCR-induced cytosolic calcium concentration changes.

## Fluctuating calcium levels are observed as a result of sequential cell contacts in vivo

We next set out to investigate if the ability of B cells to collect calcium signals sequentially is also shared by GC B cells. For two-photon intravital imaging, nitrophenyl (NP)-specific B1-8$^{hi}$ B cells from YellowCaB mice were magnetically isolated and transferred into wild-type hosts, which were subsequently immunized with NP-chicken gamma globulin (CGG) into the right foot pad (*Shih et al., 2002a*). Mice were imaged at day 8 p.i. when GCs had been fully established. Activated TN-XXL$^+$ YellowCaB cells had migrated into the GC, as confirmed by positive PNA- and anti-FP-immunofluorescence histology (*Figure 3a*). At this time point, mice were surgically prepared for imaging as described before (*Ulbricht et al., 2017*). Briefly, the right popliteal lymph node was exposed, moisturized, and flattened under a cover slip sealed against liquid drainage by an insulating compound. The temperature of the lymph node was adjusted to 37°C and monitored during the measurement. Our experiments revealed that the movement of single YellowCaB cells is traceable in vivo. Calcium fluctuations can be made visible by intensity changes in an extra channel that depicts the FRET signal, as calculated from relative quenching of TN-XXL. Color coding of intensity changes in the FRET channel showed time-dependent fluctuations of the signal and, in some particular cases, a sustained increase after prolonged contacts between two YellowCaB cells (*Figure 3b*, *Video 1*). Interestingly, FRET intensity seemed to be mostly fluctuating around low levels in moving cells, whereas sustained increase required cell arrest, as reported previously (*Negulescu et al., 1996*; *Shulman et al., 2014*), (*Figure 3—figure supplement 1*). The observed calcium fluctuations might therefore coincide with cell-to-cell contacts between follicular dendritic cells (FDCs) and B cells, resulting in AG-dependent BCR stimulation. To test for this, we first measured the colocalization between signals within the FDC channel and the citrine channel. The intensity of colocalization I$_{coloc}$ of all cells was plotted as a function of frequency and biexponentially fitted (*Figure 3c*). We set the threshold for a strong and sustained colocalization of FDCs and B cells to an intensity of 150 AU within the colocalization channel. At this value, the decay of the biexponential fit was below 10%. We thus decided to term all cells with a colocalization intensity = 0 (naturally the most abundant ones) not colocalized, cells with a colocalization intensity between 1 and 150 transiently colocalized to FDCs ('scanning' or shortly touching the FDCs), and all cells above this intensity threshold strongly or stably colocalized. We compared the relative FRET intensity changes ΔR/R of two tracked cells (*Figure 3b*, cells 1 and 2), where baseline R is the lowest FRET intensity measured, and its contacts to FDCs. We detected several transient B-cell–FDC contacts in cell 1 that were followed by a gradual increase of ΔR/R, indicating an increase of cytoplasmic calcium concentration (*Figure 3d*). Cell 2 kept strong FDC contact over the whole imaging time and maintained elevated, mostly stable values. These experiments confirmed that GC B cells are able to collect calcium as a consequence of repeated contact events, which are mediated by B cell-to-FDC contacts in vivo.

## Calibration of the TN-XXL construct for in vivo quantification of cytosolic calcium in B lymphocytes using the phasor approach to FRET–FLIM

The comparison of calcium responses in different B cell subsets of multiple GCs requires normalization of TN-XXL FRET. Since this is hardly achievable in tissue due to its inherent heterogeneity, we aimed for the determination of absolute cytosolic concentration values in YellowCaB cells by calibration of TN-XXL FRET intensities. However, ratiometric calibration would require equal conditions for

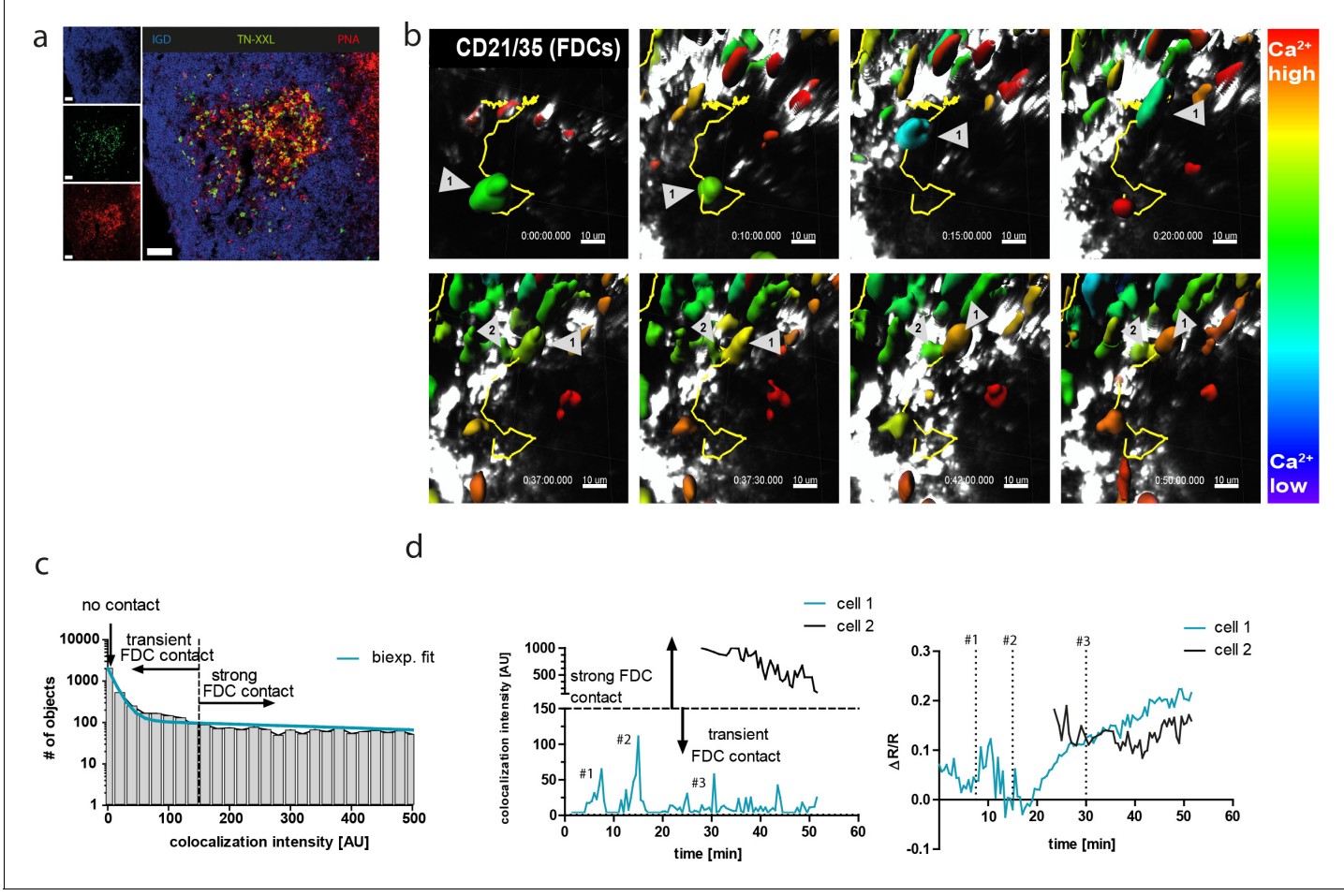

**Figure 3.** YellowCaB cells form productive germinal centers in vivo and show active B cell receptor signaling after cell-to-cell contacts. (**a**) Histological analysis of host mouse lymph nodes after adoptive transfer of YellowCaB cells. TN-XXL (green)-positive cells cluster in IgD (blue)-negative regions; a germinal center phenotype is confirmed by PNA staining (red). Scale bar 50 μm. (**b**) Stills of ratiometric intravital imaging of adoptively transferred YellowCaB cells. 3D surface rendering and single-cell tracking (track line in yellow) with relative color coding ranging from blue = low ΔR/R to red = high ΔR/R (**c**) Histogram showing segmented objects binned due to colocalization intensity within bin width of 20 AU and biexponential fit of data. Total number of objects = 6869. A curve decay of <10% was set as threshold, parting transient from strong B cell–FDC contact. All cells with colocalization intensity <1 were assigned negative. (**d**) Colocalization intensities of tracked cells 1 and 2 over time versus Förster resonance energy transfer signal change of cells 1 and 2 over time. Contact events to FDCs were assigned numbers #1, #2, and #3.

The online version of this article includes the following figure supplement(s) for figure 3:

**Figure supplement 1.** Cell velocity versus calcium flux.

donor and acceptor fluorescence signals, especially in terms of scattering and photobleaching. Due to the aforementioned heterogeneous tissue composition, these requirements cannot be met in vivo (*Radbruch et al., 2015*). Therefore, donor FLIM is the appropriate solution as it circumvents comparative evaluation of different fluorescence signals. Fluorescence lifetime is defined as the mean time a fluorescent molecule stays in an elevated energetic state after excitation, before photon emission and relaxation to the ground state take place. As a fully calcium-quenched eCFP in the GECI TN-XXL would transfer its energy mainly to citrine, its fluorescence lifetime would be measurably shorter than that of unquenched eCFP. Time-correlated single-photon counting (TCSPC) devices offer the possibility of simultaneous photon detection and recording of the respective fluorescent lifetimes within a nanosecond scale, yielding a fluorescence lifetime decay histogram of photons. Deriving fluorescent lifetimes τ from the eCFP decay curve of photon histograms requires fitting. We decided to use the phasor analysis as it virtually transfers time-resolved fluorescence data into phase domain data by discrete Fourier transformation (*Digman et al., 2008*). This approach overcomes the

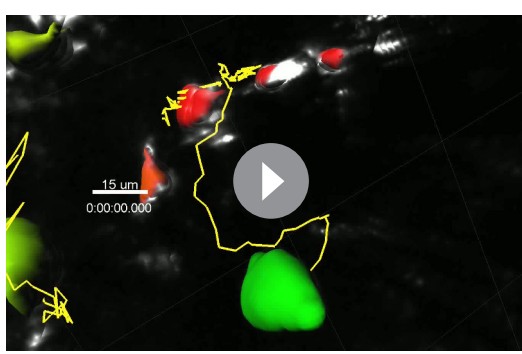

**Video 1.** Detail of intravital ratiometric imaging (day 8 p.i.) within germinal center. YellowCaB cells had been adoptively transferred, and FDCs were in vivo-labeled with anti-CD21/35 antibody (white). 3D surface rendering and single-cell tracking (track line in yellow) with color coding ranging from blue = low ΔR/R to red = high ΔR/R. 103 frames, 7 frames per second (fps), scale bar 50 μm.
https://elifesciences.org/articles/56020#video1

obstacles of multicomponent exponential analysis and yields model-free, readily comparable pixel- or cell-based plots that assign a position within a semicircle to each data point, dependent on the mixture of lifetime components present (*Leben et al., 2018*).

Typically, data correction based on reference dyes is needed for reliable phasor analysis (*Ranjit et al., 2018*). We verified the reliability of our TCSPC setup to acquire high-quality fluorescence decays in an image to be evaluated using the phasor approach by measuring the instrument response function given by the second harmonic generation signal (SHG) of potassium-dihydro-phosphate crystals (laser excitation wavelength 940 nm) and the fluorescence decays of eGFP, expressed in HEK cells (laser excitation wavelength 900 nm). As shown with the phasor plots of the raw data (*Figure 4a*), both the SHG signal and the eGFP fluorescence are located in expected positions on the semicircle (*Murakoshi et al., 2008*; *Rinnenthal et al., 2013*). Therefore, no further correction of the data is necessary in our system.

As the TN-XXL construct is exclusively expressed in the cytosol of B lymphocytes from YellowCaB mice, the following equilibrium holds true for $Ca^{2+}$ (free cytosolic calcium), *TNXXL* (the calcium-free FRET construct, i.e., the unfolded tertiary structure of TnC), and $Ca^{2+}TNXXL$ (the FRET construct saturated by calcium, i.e., the completely folded tertiary structure of TnC):

$$Ca^{2+} + TNXXL \rightleftharpoons Ca^{2+}TNXXL \tag{1}$$

characterized by the dissociation constant $K_d$:

$$K_d = \frac{[Ca^{2+}TNXXL]}{[Ca^{2+}][TNXXL]} \tag{2}$$

As measured in lysate of YellowCaB B cells expressing TN-XXL, the fluorescence lifetime τ of the donor eCFP of the FRET Ca-sensitive construct TN-XXL depends on the free calcium concentration [Ca²⁺] following a sigmoidal function (Eq. 3).

$$\tau = \tau_{FRET} + \frac{\tau_{free} - \tau_{FRET}}{1 + 10^{-(log_{10}[Ca^{2+}] - log_{10}K_d) \cdot Hill\_slope}} \tag{3}$$

with τ_free the fluorescence lifetime eCFP at 0 nM free calcium, and τ_FRET the fluorescence lifetime of completely FRET-quenched eCFP in the TN XXL-construct at 39 μM free calcium. By fitting the fluorescence lifetime of eCFP in TN-XXL excited at 850 nm and detected at 460 ± 30 nm acquired in time domain using our TCSPC system at various free calcium concentrations, we determined $K_d$ = 475 ± 46 nM and Hill slope = −1.43 ± 0.17 (*Figure 4b*). Thus, we can calculate the free calcium concentration as

$$log_{10}[Ca^{2+}] = log_{10}K_d - \frac{log_{10}\left(\frac{\tau_{free} - \tau_{FRET}}{\tau - \tau_{FRET}} - 1\right)}{Hill\_slope} \tag{4}$$

Similar to the calculation in time domain, for the phase domain we can express each phasor vector $\vec{p}$ based on the formalism proposed by Celli and colleagues (*Celli et al., 2010*) as

$$\vec{p} = \frac{[Ca^{2+}\ TNXXL]\varepsilon_{FRET}\vec{p}_{FRET} + [TNXXL]\varepsilon_{free}\vec{p}_{free}}{[Ca^{2+}\ TNXXL]\varepsilon_{FRET} + [TNXXL]\varepsilon_{free}} \tag{5}$$

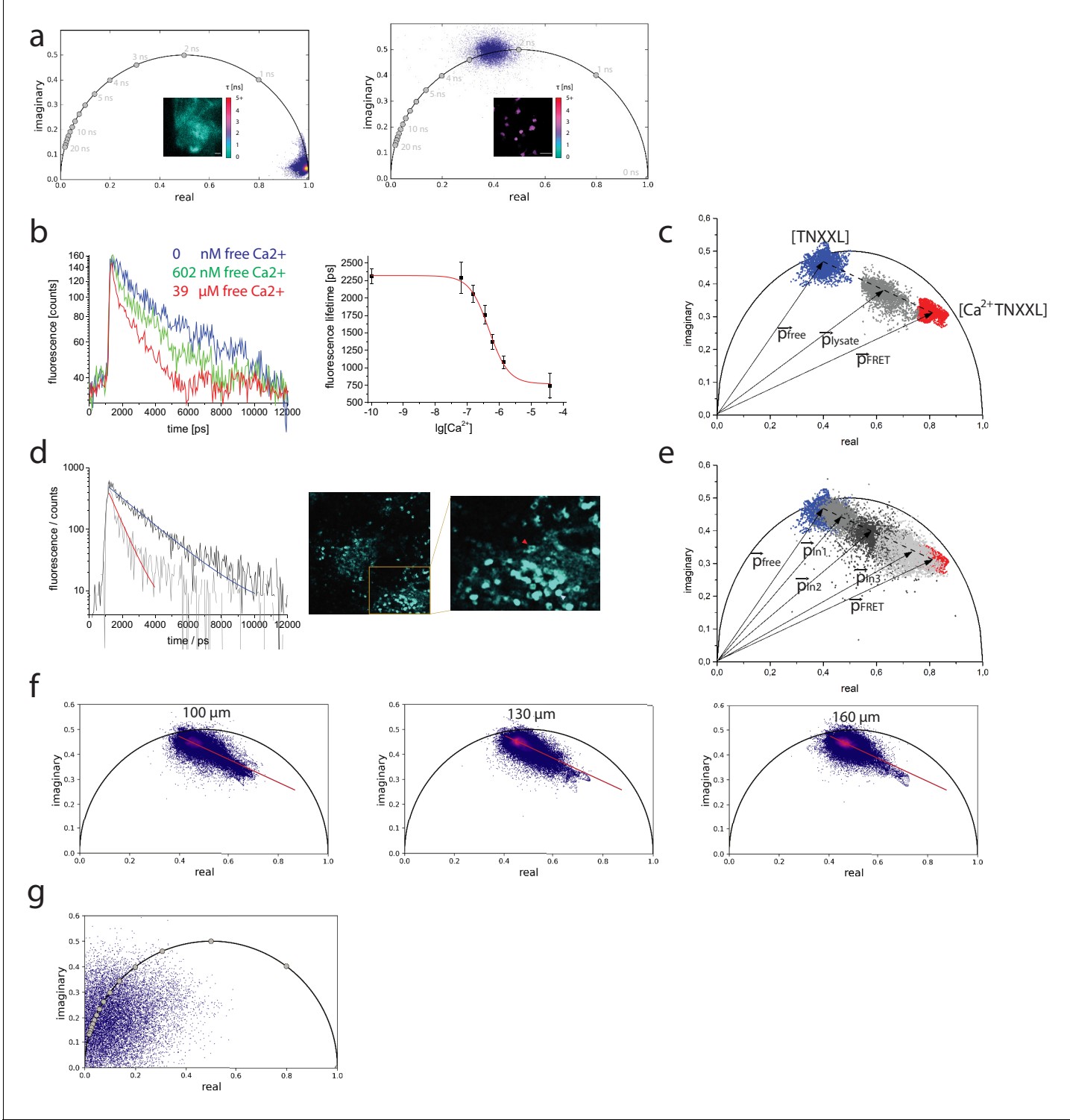

**Figure 4.** Calibration of the TN-XXL construct using fluorescence lifetime imaging of its Förster resonance energy transfer donor. (**a**) Left panel: phasor plot of second harmonic generation signal of potassium dihydrogen phosphate crystals and lifetime image (inset, scale in picoseconds) corresponding to the instrument response function ($\tau$ = 80 ± 10 ps). $\lambda_{exc}$ = 940 nm, $\lambda_{detection}$ = 466±20 nm. Right panel: phasor plot of GFP fluorescence expressed in HEK cells and fluorescence lifetime image (inset, the same scale as in the left panel), corresponding to mono-exponential decay GFP fluorescence ($\tau$ = 2500 ± 100 ps). $\lambda_{exc}$ = 900 nm, $\lambda_{detection}$ = 525±25 nm. (**b**) Left panel: fluorescence decays of CFP in TN-XXL construct from lysates of B lymphocytes at 0 nM, 602 nM, and 39 μM free calcium. Right panel: titration curve of TN-XXL resulting from the time-domain evaluation of decay curves as shown in the left panel (three independent experiments). $\lambda_{exc}$ = 850 nm, $\lambda_{detection}$ = 466±20 nm. (**c**) Phasor plot of representative data shown in (**b**) – time-

*Figure 4 continued on next page*

*Figure 4 continued*

resolved fluorescence images 422 × 422 pixels (200 × 200 µm²); time-bin = 55 ps; time window = 12.4 ns. Blue phasor cloud (with central phase vector $\vec{p}_{free}$) corresponds to 0 nM free calcium, gray cloud (with central phase vector $\vec{p}_{lysate}$) to 602 nM free calcium, and red cloud (with central phase vector $\vec{p}_{FRET}$) to 39 µM free calcium. The dotted line connects the centers of the blue and red clouds, respectively, whereas the gray cloud is located on this line. The dotted line corresponds to the calibration segment as it results from measurements of TN-XXL in cell lysates. (d) Left panel: representative fluorescence decays of eCFP in two B lymphocytes (indicated by red and blue arrowheads in the inset image, right panel) expressing TN-XXL in the medullary cords of a popliteal lymph node of a YellowCaB mouse (right panel) and corresponding mono-exponential fitting curves (red fitting curve: τ = 703 ± 56 ps; blue fitting curve: τ = 1937 ± 49 ps). We measured τ = 2303 ± 53 ps in splenocytes expressing only CFP. (e) Phasor plot showing time-resolved CFP fluorescence data from three lymph nodes, in three YellowCaB mice (light gray – with central phase vector $\vec{p}_{ln3}$, gray – with central phase vector p $\vec{p}_{ln1}$, and dark gray – with central phase vector $\vec{p}_{ln2}$) – time-resolved fluorescence images 505 × 505 pixels (512 × 512 µm²); time-bin = 55 ps; time window = 12.4 ns. Additionally, the calibration segment (dotted line) and the phasor clouds measured in lysates of B lymphocytes expressing TN-XXL at 0 nM and 39 µM free calcium from (c) are displayed. (f) Phasor plots of the CFP fluorescence (time-resolved fluorescence images 505 × 505 pixels / 512 × 512 µm²) acquired at three different depths (100, 130, and 160 µm from the organ capsule surface) in the popliteal lymph node of a YellowCaB mouse. The red line in each phasor plot represents the calibration segment also displayed in (c) and (e). (g) Phasor plot of signal acquired in the lymph node of a non-fluorescent mouse. λ_exc = 850 nm, λ_detection = 466±20 nm.
The online version of this article includes the following figure supplement(s) for figure 4:

**Figure supplement 1.** Phasor plot showing time-resolved CFP fluorescence data of B lymphocytes from YellowCaB mice in culture (gray cloud – with central phase vector $\vec{p}_{Bcells}$) – time-resolved fluorescence images 471 × 471 pixels (250 × 250 µm²); time-bin = 55 ps; time window = 12.4 ns.

with $\varepsilon_{free}$ and $\varepsilon_{FRET}$ the relative brightness (*Chen et al., 1999*; *Müller et al., 2000*) of eCFP in TNXXL at 0 nM and saturated (39 µM) free calcium, given by the following equations:

$$\varepsilon_{free} = \delta_{CFP} \cdot \eta_{free} = \delta_{CFP} \cdot k_F \cdot \tau_{free} \tag{6}$$

$$\varepsilon_{FRET} = \delta_{CFP} \cdot \eta_{FRET} = \delta_{CFP} \cdot k_F \cdot \tau_{FRET} \tag{7}$$

with $\delta_{CFP}$ the effective two-photon absorption cross section of eCFP (independent of the pathways of relaxation from the excited state), and $k_F$ the fluorescence rate of eCFP in vacuum, that is, no quenching due to surrounding molecules.

The phase vectors can be written also as complex numbers as given by the following equations:

$$\vec{p} = Re + i \cdot Im \tag{8}$$

$$\vec{p}_{free} = Re_{free} + i \cdot Im_{free} \tag{9}$$

$$\vec{p}_{FRET} = Re_{FRET} + i \cdot Im_{FRET} \tag{10}$$

We determined the averages and median real and imaginary values of the phasor distributions obtained by performing FRET–FLIM in lysates of YellowCaB B cells at 0 nM and 39 µM free calcium to be $Re_{free}$ = 0.4035 (average), 0.40326 (median); $Im_{free}$ = 0.45801 (average), 0.45779 (median) and $Re_{FRET}$ = 0.82377 (average), 0.82203 (median); $Im_{FRET}$ = 0.3225 (average), 0.32093 (median), respectively, indicating that both distributions are symmetric, corresponding to normal distributions (*Figure 4c*).

From Eqs. (5–10) combined with Eq. (2), the free cytosolic calcium concentration is given by

$$\left[Ca^{2+}\right] = K_d \frac{\varepsilon_{free}}{\varepsilon_{FRET}} \cdot \frac{\vec{p} - \vec{p}_{free}}{\vec{p}_{FRET} - \vec{p}} = K_d \frac{\tau_{free}}{\tau_{FRET}} \cdot \frac{\sqrt{\left(Re - Re_{free}\right)^2 + \left(Im - Im_{free}\right)^2}}{\sqrt{\left(Re_{FRET} - Re\right)^2 + \left(Im_{FRET} - Im\right)^2}} \tag{11}$$

Thus, the free calcium concentration depends only on the $K_d$, $t_{free}$, $t_{FRET}$ as determined from the calibration curve measured in lysate and the phase vectors, describing the extreme states of eCFP in the TN-XXL construct.

Since variations in refractive index, ion strength, *pH* value, or temperature in the cytosol of the B lymphocytes may additionally influence the fluorescence lifetime of eCFP, as well as the phase

vectors $\vec{p}_{free}$ and $\vec{p}_{FRET}$ (*Jameson et al., 1984*; *Scott et al., 1970*), we verified whether the FRET trajectory of the-TN-XXL construct changes in the cytosol of B lymphocytes in cell culture (*Figure 4—figure supplement 1*) and under in vivo conditions, in lymph nodes (*Figure 4d, e*). We found that in all our measurements the phasor cloud lays on the trajectory determined in lysate solutions (*Figure 4e*). Measurements performed at different depths in lymph nodes led to the same result: there is no change in the slope of the trajectory at different tissue depths (*Figure 4f*).

To assess the impact of autofluorescence on the interpretation of the fluorescence signal in the phasor plot and, thus, on the cytosolic calcium levels, we also performed FLIM in B cell follicles of lymph nodes of non-fluorescent wild-type mice. While the acquired signal was extremely low, the phasor cloud in these measurements was located around position (0,0) in the plot, indicating that it mainly originates from detector noise (*Figure 4g*).

We compared the results of cytosolic free calcium concentration determined using the time-domain and the phase-domain approach and found deviations of max. 5% between the evaluation pathways using Eqs. (4) and (11) due to numerical uncertainty. We determined the calcium dynamic range of TN-XXL measured by phasor-analyzed FRET–FLIM to span between 100 nM and 4 µM free calcium, with the linear range of the titration curve in the range between 265 and 857 nM free calcium.

## Comparative FLIM–FRET reveals heterogeneity of absolute calcium concentrations in B cells in vivo

Adoptively transferring AG-specific YellowCaB cells and non-AG-specific (polyclonal) YellowCaB cells (stained ex vivo) allowed us to divide GC B cells into five different populations based on their location in the imaging volume and their fluorescent appearance (*Figure 5a, b*). At day 8 p.i., polyclonal YellowCaB cells, identified by their red labeling, mostly lined up at the follicular mantle around the GC, with some of them having already entered into activated B cell follicles. AG-specific, citrine-positive B1-8[hi]:YellowCaB cells were found clustered in the GC, close to FDCs, outside of GC boundaries, or as bigger, ellipsoid cells in the extrafollicular medullary cords (MC), probably comprising plasma blasts (*Figure 5b*, left). Color-coded 2D- and 3D FLIM analysis of these populations confirmed that calcium concentrations were fluctuating within all of those B cell populations, and that most B cells were maintaining relatively high mean eCFP fluorescence lifetimes and therefore low calcium concentrations on average, with only few exceptions (*Figure 5b*, middle and right, *Video 2*).

Bulk analysis of cells revealed additional calcium-intermediate and calcium-high cell subsets present among AG-specific cells and plasma blasts (*Figure 5c*, *Video 3*). Especially for plasma blasts this was somehow unexpected, given that these are thought to downregulate their surface BCR during differentiation.

Comparison of AG-specific cells inside GCs with those outside GCs and non-AG-specific cells inside GCs as well with those outside GCs showed that the distribution of calcium concentrations of these B cells was dependent on BCR specificity and rather independent from their location within the imaging volume, despite higher fluctuation seen among AG-specific populations (*Figure 5—figure supplement 1*). These maxima were reached as transient fluctuation peaks, that is, periods shorter than 1 min, in which these concentrations seem to be tolerated. Calcium values exceeding the dynamic range of TN-XXL (>857 nM) were recorded for all measured subsets, but most cells >857 nM were found among intrafollicular AG-specific B cells and extrafollicular AG-specific B cells (*Figure 5d*). The heterogeneity in temporal calcium concentrations therefore is smallest among non-AG-specific B cells, increases with activation in AG-specific GC B cells, and is most prominent among MC-plasma blasts. The high cytoplasmic calcium levels observed in MC-plasma blasts were unlikely to be the result of chemokine-induced signaling since only a minor calcium increase was detected after CXCL12 stimulation in vitro (*Figure 2—figure supplement 1d*). Thus, a progressive heterogeneity of calcium signals within B cells can be seen alongside the process of activation and differentiation.

## Functional relevance of increased calcium concentration among extrafollicular YellowCaB cells

We next wondered if high cytoplasmic calcium levels within certain B cell subsets could be the result of AG-mediated signals. We therefore intravenously (i.v.) injected NP-bovine serum albumine (BSA)

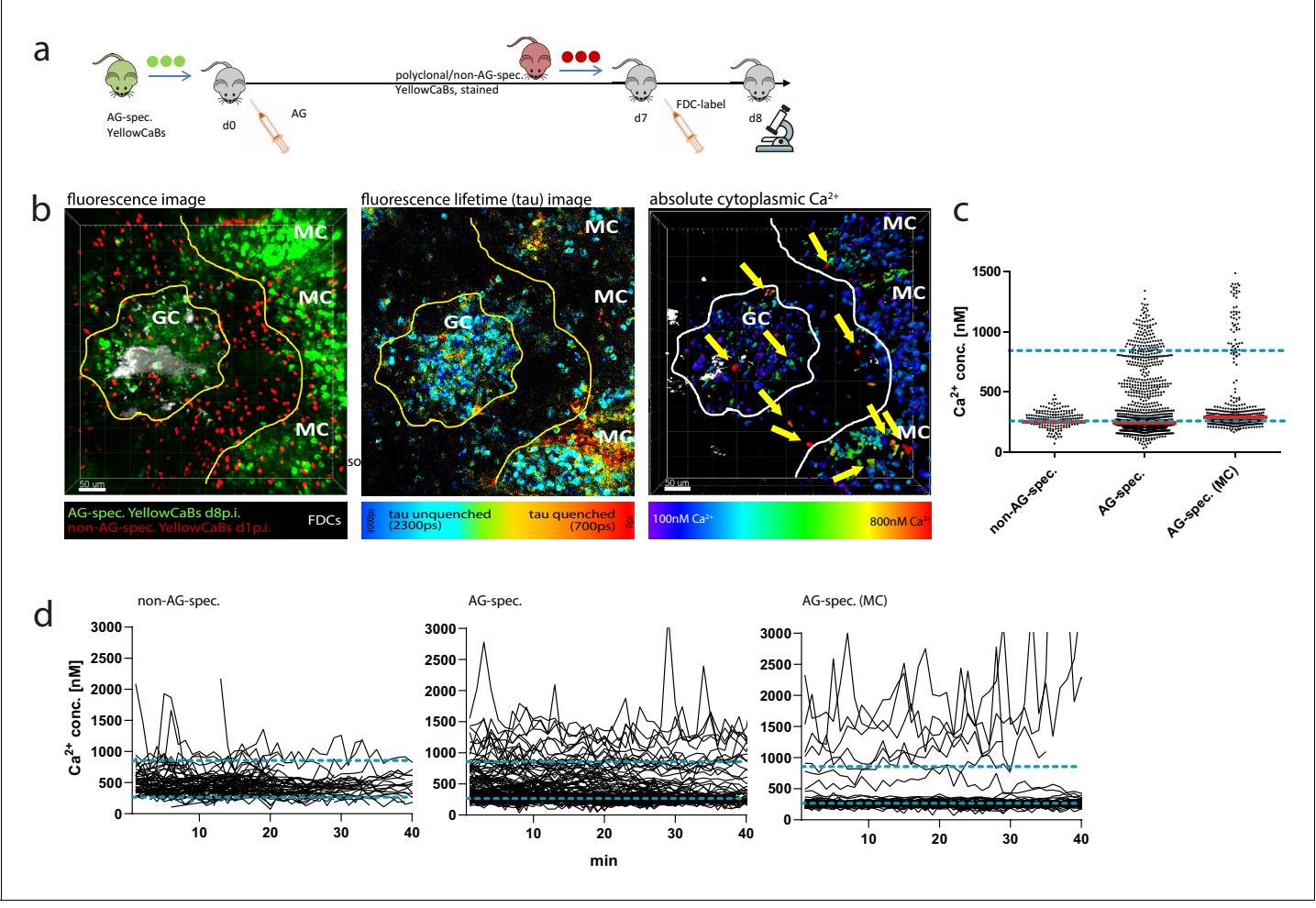

**Figure 5.** Determination of absolute calcium concentration by intravital fluorescence lifetime imaging of germinal center (GC) B cell populations. (a) Cell transfer and immunization strategy for intravital imaging of antigen (AG)-specific and polyclonal YellowCaB cells. (b) Left: maximum intensity projection of a z-stack, intravitally imaged GC, and medullary cords (MCs). B cells were distinguished as polyclonal, non-AG-specific YellowCaB cells (red), AG-specific YellowCaB cells, and AG-specific cells inside the MC. Middle: color-coded fluorescence lifetime image with lifetimes of unquenched eCFP depicted in blue and lifetimes of quenched eCFP in red. Right: 3D-rendered, color-coded z-stack showing absolute calcium concentrations in GC and MC. Yellow arrows point to cells containing high cytoplasmic calcium. (c) Bulk analysis of absolute calcium concentrations in segmented single-cell objects from B cell subsets at any given time point measured. The dynamic range of the genetically encoded calcium indicator (GECI) TN-XXL is indicated by blue dashed lines. (d) Time-resolved analysis of calcium concentrations in tracked segmented objects corresponding to B cell subsets in (c). 2 frames per minute. Non-AG-specific YellowCaB cells (left, n = 92 tracks); AG-specific YellowCaB cells (middle, n = 169 tracks); and extrafollicular AG-specific YellowCaB cells in MC (right, n = 69 tracks). The dynamic range of the GECI TN-XXL is indicated by blue dashed lines.

The online version of this article includes the following figure supplement(s) for figure 5:

**Figure supplement 1.** Mean calcium concentration and SD in non-antigen (AG)-specific and AG-specific YellowCaB cells distinguished by localization.

(66 kDa) into mice that had been adoptively transferred with B1-8$^{hi}$ (AG-specific) YellowCaB cells and recorded absolute calcium concentrations within the GC by intravital FLIM. Antigens up to 70 kDa have been reported to be transported into the follicles via conduits in less than 5 min (*Roozendaal et al., 2009*). Accordingly, AG-specific GC YellowCaB cells significantly upregulated cytoplasmic calcium after AG injection within minutes (*Figure 6a*, top panel, *Figure 6—figure supplement 1*). To test if this concentration increase could be explained via BCR-dependent AG recognition, we pre-injected the inhibitor of Bruton's tyrosine kinase (BTK) ibrutinib (*Hendriks et al., 2014*), which blocks BCR downstream activation, in a control group. BTK inhibition could abrogate the increase in mean cytoplasmic calcium after additional injection of NP-BSA (*Figure 6a*, bottom panel) and even seemed to downregulate baseline signals (*Figure 6—figure supplement 1*). In addition, we found that also in 48 hr LPS/ interleukin 4 (IL-4)-cultured B1-8-plasma

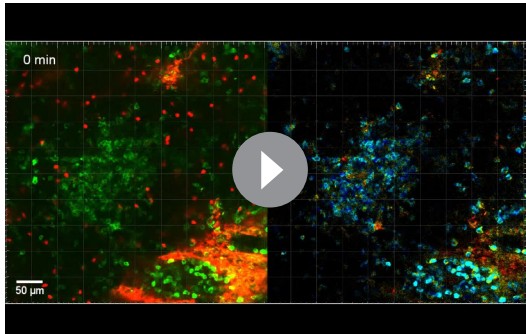

**Video 2.** Side-by-side depiction of fluorescence, fluorescence lifetime imaging, and cell-based phasor data of intravitally imaged germinal center (day 8 p.i.), single z-plane. Left: fluorescence data with antigen (AG)-specific YellowCaB cells (green) and stained non-AG-specific YellowCaB cells transferred 1 day prior to imaging (red; autofluorescence of capsule also visible in the same channel). 4 fps, scale bar 50 μm. Middle: false color-coded presentation of fluorescence lifetime τ (0–3000 ps, see range scale in *Figure 5b*). 4 fps, scale bar 50 μm. Right: raw cell-based phasor plot with cells segmented according to fluorescence and spatial distribution, subsets indicated. 4 fps.

https://elifesciences.org/articles/56020#video2

blasts, an increase in calcium was detectable after addition of NP-BSA (*Figure 6—figure supplement 2*), suggesting that stimulation via AG remains possible in at least a proportion of these differentiated B cells.

These results led us to investigate the possible sources of AG abundance outside of GCs and their effect on calcium in B cells. Earlier studies proposed that one possible AG source in lymph nodes are subcapsular sinus macrophages (SCSM) (*Junt et al., 2007*; *Moran et al., 2018*; *von Andrian and Mempel, 2003*). We tested if SCSM contacts could be the cause of elevated calcium levels in extrafollicular B cells. We intravitally imaged wild-type host mice that have been adoptively transferred with B1-8hi:YellowCaB cells and received an injection of efluor660-labeled anti-CD169 antibody together with the usual FDC labeling 1 day prior to analysis. We concentrated on the area beneath the capsule, identified by second harmonic generation signals of collagen fibers in this area. Thresholds of colocalization between CD169[+] macrophages and TN-XXL[+] YellowCaB cells are described in *Figure 6—figure supplement 3*. Together, these methods led to a 3D visualization of the SCS with CD169 stained macrophages, lined up in close proximity (*Figure 6b*). AG-specific YellowCaB cells were detected clustering in GCs nearby. Extrafollicular

YellowCaB cells crowding the SCS space were found to have multiple contact sites to SCSM. Some B cells were observed to migrate along the SCS, possibly scanning for antigenic signals (*Video 4*). Bulk analysis of YellowCaB cells and their colocalization with SCSM showed that the calcium concentration in YellowCaB cells with direct contact to SCSM reaches values that are more than doubled compared to values in cells that were not in contact, and that calcium concentration is positively cor-

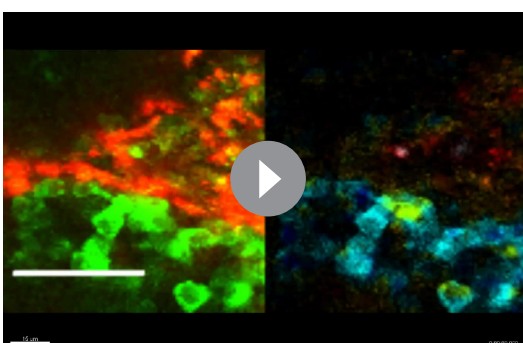

**Video 3.** Detail of *Video 2* within medullary cords and side-by-side depiction of fluorescence and fluorescence lifetime imaging data, single z-plane. Left: fluorescence data with antigen-specific YellowCaB cells (green) and autofluorescence of capsule (red). 4 fps, scale bar 50 μm. Right: false color-coded presentation of fluorescence lifetime τ (0–3000 ps, see range scale in *Figure 5b*). 4 fps.

https://elifesciences.org/articles/56020#video3

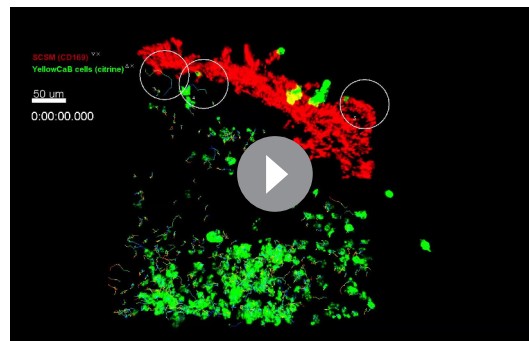

**Video 4.** 3D projection of intravital imaging of germinal center and subcapsular sinus. YellowCaB cells (green, with track lines) and subcapsular sinus macrophages (red), stained by CD169 in vivo labeling. White circles highlight antigen-specific B cells migrating along subcapsular space. 4 fps, scale bar 50 μm.

https://elifesciences.org/articles/56020#video4

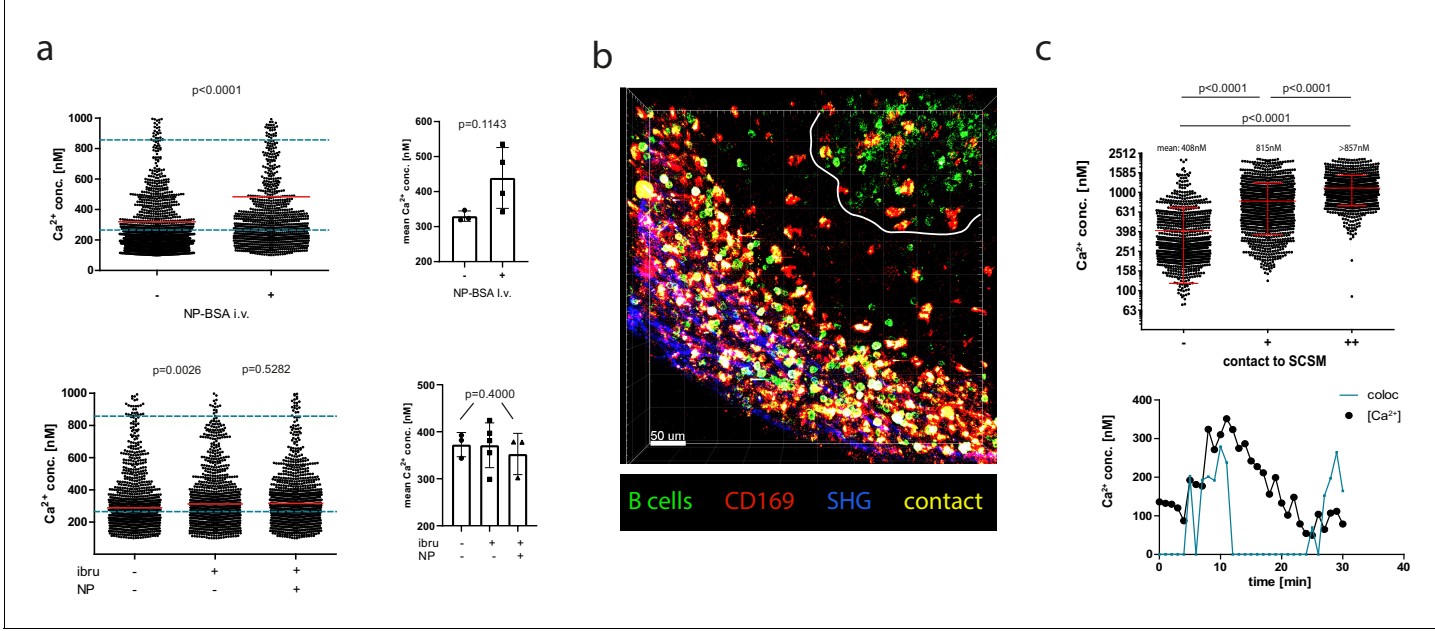

**Figure 6.** Antigen dependency of calcium elevation in germinal center (GC) and extrafollicular B cells. (a) Top panel: absolute calcium concentrations measured in antigen (AG)-specific GC B cells before and after in vivo injection of NP-BSA. Exemplary results (left) and pooled data from three imaged mice (right). Bottom panel: absolute calcium concentrations measured in AG-specific GC B cells before and after in vivo injection of the Bruton's tyrosine kinaseinhibitor ibrutinib, followed by injection of NP-BSA. Exemplary results (left) and pooled data from three imaged mice (right). (b) z-stack of intravitally imaged lymph node with GC (white line) and subcapsular sinus (indicated by SHG, blue). CD169[+] macrophages (red, contacts [yellow], YellowCaB cells [green]). Size 500 × 500 × 78 µm. Scale bar 60 µm. (c) Top: Fluorescence lifetime imaging measurement of mean absolute calcium concentration of YellowCaB cells showing no (–), transient (+), or strong (++) overlap with CD169[+] signal. n = 1000, ANOVA analysis, mean and SD. Bottom: Single-cell track of a YellowCaB cell making transient contact to a macrophage; blue: colocalization intensity (AU); black: change of absolute calcium concentration.

The online version of this article includes the following figure supplement(s) for figure 6:

**Figure supplement 1.** Calcium concentration change detected by in vivo fluorescence lifetime imaging measurements over time, exemplary single germinal center (GC) B cell tracks, before and after injection(s) of compounds.

**Figure supplement 2.** Absolute calcium concentration of fluorescence lifetime imaging measured after NP-BSA stimulation of ex vivo lipopolysaccharide-induced plasmablasts.

**Figure supplement 3.** Colocalization histogram and exponential fit for analysis of colocalization between CD169[+] macrophages and extrafollicular YellowCaB cells.

related with contact strength (*Figure 6c*, top). Single-cell tracking and simultaneous analysis of absolute calcium concentration and colocalization intensity revealed that the increase of cytoplasmic calcium is a direct cause of B-cell-to-SCSM contacts (*Figure 6c*, bottom). We conclude that contacts of B cells to SCSM could induce elevation of B cell cytoplasmic calcium concentrations, presumably due to antigenic activation, with the absolute concentrations being dependent on the contact strength.

## Discussion

Intravital imaging technologies have contributed greatly to a better understanding of the dynamic processes in GCs. AG-capture, cycling between zones, and development of clonality patterns have been made visible by two-photon microscopic techniques (*Hauser et al., 2007*; *Suzuki et al., 2009*; *Tas et al., 2016*; *Victora et al., 2010*). Furthermore, important functional in vivo data like signaling in T helper cells have been collected using a calcium-sensitive protein (*Kyratsous et al., 2017*; *Shulman et al., 2014*).

However, calcium mobilization within GC B cells was mostly investigated via ex vivo analysis of sorted cells after adoptive transfer and immunization, or BCR activation was measured using a non-reversible BCR signaling reporter like Nur77, altogether neglecting calcium flux (*Khalil et al., 2012*; *Mueller et al., 2015*). These data suggested that BCR signaling in the GC is reduced. However, no statement was made about the dynamics and timely coordination of (even small) calcium pulses and the relation to their microenvironment. In fact, a recent study confirmed that BCR signals play a central role in positive selection and display a fragile interdependence with costimulatory events (*Luo et al., 2018*).

BCR-regulating surface proteins like CD22 or sialic acid-binding immunoglobulin-type lectins have been related to development of autoimmunity and point out BCR-mediated calcium flux as an important component, not only during B cell development but also in their differentiation to effector cells (*Hoffmann et al., 2007*; *Jellusova et al., 2010*; *Müller and Nitschke, 2014*; *Nitschke and Tsubata, 2004*; *O'Keefe et al., 1999*). Apart from regulating gene transcription, cytosolic calcium mobilized has been shown to be essential for F-actin remodulation and B cell spreading on antigen presenting cells (*Maus et al., 2013*). Furthermore, cytosolic calcium concentration is closely linked to metabolic reprogramming of activated B cells and their cell fate (*Boothby and Rickert, 2017*; *Caro-Maldonado et al., 2014*). It has been shown that SOCE is acting directly on the mitochondrial capability to import cytosolic calcium (*Shanmughapriya et al., 2016*). In mitochondria, calcium is regulating ATP production through increase of glycolysis and fatty acid oxidation, processes for which activated and GC B cells have high demands, although there is controversy as to which of the two metabolic pathways predominates in GC B cells (*Griffiths and Rutter, 2009*; *Jellusova et al., 2017*; *Maus et al., 2017*; *Weisel et al., 2020*).

For flexible analysis of calcium mobilization in cells of the $CD19^+$ lineage, we developed a novel transgenic reporter system and image processing approach, enabling quantification of cytosolic calcium concentrations. The FRET-based GECI TN-XXL can be used stably in moving, proliferating, and differentiating lymphocytes, and the reversibility of the sensor makes it suitable for longitudinal intravital measurements. Switching from ratiometric acquisition of FRET-donor and FRET-acceptor fluorescence intensities to measuring FRET-donor fluorescence lifetime enabled quantification of calcium concentrations in absolute numbers.

A first advantage of FRET–FLIM in tissue is that different photobleaching or scattering properties of the fluorophores can be neglected. We further decided to perform all analyses based on the phasor approach that circumvents the problem of multiexponential fluorescence decays we encounter measuring a two-fluorophore FRET-based GECI in tissue. For titration of TN-XXL, we used lysate of YellowCaB plasma blasts. Besides TN-XXL affinity, also eCFP fluorescence lifetime itself may be influenced by large shifts in *pH* value, ionic strength, oxygenation, or temperature. We ensured that these parameters were similar in lysates and cells, except for the temperature, which was at room temperature for calibration. Temperature was reported to only slightly change the fluorescence lifetime of a CFP variant (cerulean) (*Laine et al., 2012*). However, for accuracy purposes and in order to exclude such artifacts when determining cytosolic calcium levels in B lymphocytes within lymph nodes of YellowCaB mice, we adapted the phasor-based calibration strategy proposed by *Celli et al., 2010* for the use of Calcium Green in skin samples to our data. In this way, we were able to reliably determine absolute values of cytosolic calcium concentrations in B cells within lymph nodes.

In our set-up, we have shown that TN-XXL in B cells has suitable sensitivity and fast reversibility. This key factor made it possible to observe repeated and partially sustained calcium elevation in the cytoplasm, showing that B cells are able to collect sequential signals, possibly up to a certain threshold, which determines their fate.

In support of that, B cellular calcium concentration must not constitutively exceed a certain value in order to prevent mitochondrial depolarization (*Akkaya et al., 2018*; *Bouchon et al., 2000*; *Niiro and Clark, 2002*). Gradual calcium elevation could be a mechanistic link for that. For example, calcium levels of >1 µM over the duration of >1 hr have been reported to be damaging to other cell types, such as neurons (*Radbruch et al., 2015*; *Siffrin et al., 2015*). Accordingly, stimulation of AG receptors via large doses of soluble AG can lead to tolerogenic apoptosis in GC B cells, which could be explained by uninhibited calcium influx (*Nossal et al., 1993*; *Pulendran et al., 1995*). Since apoptosis is the default fate for B cells in the GC reaction (*Mayer et al., 2017*), CD40 and TLR signaling might contribute to limiting cytoplasmic calcium concentrations, and thus promote survival of B cell

clones with appropriate BCR affinity (*Akkaya et al., 2018*; *Eckl-Dorna and Batista, 2009*; *Pone et al., 2015*; *Pone et al., 2012*; *Pone et al., 2010*; *Ruprecht and Lanzavecchia, 2006*). For CD40 signaling in immature B cells, this has been confirmed (*Nguyen et al., 2011*). Our data does show that TLR signaling can attenuate calcium flux in stimulated B cells, while CD40 can either attenuate or augment the calcium response (*Figure 2*). Whether the different outcomes of CD40 stimulation are dependent on the affinity of the BCR and its efficiency in presenting AG (*Schwickert et al., 2011*; *Shulman et al., 2013*) will be subject of further studies.

Measuring absolute calcium concentration in GC B cells after administration of soluble AG in vivo, we could detect an increase of B cell calcium that is attenuated by BCR inhibitor ibrutinib, showing that BCR-mediated calcium increase is substantially contributing to calcium heterogeneity in the GC. However, the interpretation of the data should not neglect other causes of calcium elevation, given the multifaceted role of this second messenger. Therefore, it is likely that apart from BCR signaling, also other events, like binding of non-AG ligands and stress-related calcium release from internal stores, contribute to an overall cytosolic calcium increase, which needs to be regulated in order to prevent a damaging calcium overload. Causes for stress-related cytosolic calcium elevations in cells can be hypoxia, a condition reported to be present within GCs (*Jellusova et al., 2017*); nutrient deprivation, which mostly will affect highly proliferative cells like GC B cells; or ER-calcium release as a result of the unfolded protein response that is indispensable in plasma cells (*Díaz-Bulnes et al., 2020*; *Høyer-Hansen and Jäättelä, 2007*; *Lam and Bhattacharya, 2018*). The complex interaction of factors makes an exact characterization of the absolute calcium concentration in various B cell subsets crucial in order to obtain information about their regulation and containment. This characterization should preferably be done intravitally since any manipulation of cells can result in enormous non-physiological variations of cytosolic calcium levels.

For the first time, we determined absolute values of B cell cytoplasmic calcium concentrations during the GC reaction within living mice. It appears that BCR AG specificity and state of differentiation are closely related to distinct degrees of heterogeneity of calcium concentrations. Notably, heterogeneity was also evident in extrafollicular B cells in the SCS region, as well as in plasma blasts. The latter actually reach the highest calcium concentrations within the B cell compartment of the lymph node. We also observed an increase of cytoplasmic calcium in plasma blasts after stimulation with specific AG in vitro. These data are in line with a report of residual BCR signaling occurring in antibody-secreting cells (*Pinto et al., 2013*), which challenges the finding that B cells completely downregulate their surface BCR during differentiation to plasma cells (*Manz et al., 1998*). Our experiments were done in short-lived plasma blasts, not LLPC, for which the situation could be different. Therefore, an investigation of possible BCR signaling in LLPC is of high interest. Stimulation with the chemokine CXCL12, which has previously been shown to induce migration of antibody-secreting cells (*Fooksman et al., 2010*; *Hauser et al., 2002*), resulted only in a minor increase of cytoplasmic calcium in plasma blasts in our hands.

In B cells that have exited the GC, ongoing calcium flux might reflect reactivation. We confirmed that B cells in contact to SCSM had significantly higher cytosolic calcium concentrations. These are possibly attributed to BCR signaling since the SCS has been proposed as a site of reactivation of memory B cells via AG (*Moran et al., 2018*).

The YellowCaB system provides a tool for measuring calcium as ubiquitous, universal cellular messenger, integrating signals from various pathways, including chemokine receptor signaling and intrinsic calcium release or BCR-triggered activation. Importantly, changes in mitochondrial membrane potential and/or the integrity of the ER also lead to varying calcium concentrations within the cytoplasm since both act as major intracellular calcium buffering organelles (*Kass and Orrenius, 1999*) A close connection between mitochondrial calcium homeostasis, altered reactive oxygen speciesproduction, and the expression of plasma cell master transcription factor BLIMP1, as well as changes in metabolism, has been reported previously (*Jang et al., 2015*; *Shanmugapriya et al., 2019*). We have recently applied phasor-FLIM of endogenous NAD(P)H fluorescence for mapping of metabolic enzyme activities in cell cultures (*Leben et al., 2019*). The combination of this technique with FLIM-based intravital calcium analysis will help to further dissect immunometabolic processes in B cells, as well as in short-lived plasma cells and LLPCs in vivo.

# Materials and methods

## Key resources table

| Reagent type (species) or resource | Designation | Source or reference | Identifiers | Additional information |
|---|---|---|---|---|
| Mouse | R26CAG-TNXXLflox | *Mank et al., 2008* | - | Plasmid available at addgene (#45797) |
| Mouse | CD19 cre | *Rickert et al., 1997* | Stock# 004126 jaxmice | |
| Mouse | B1-8$^{hi}$gh | *Shih et al., 2002a* | Stock# 007775 jaxmice | |
| Antibody | Rabbit anti-GFP-Alexa488 polyclonal | Rockland, in-house coupling DRFZ Berlin | Cat# 600-401-215 | 1:200 |
| Antibody | Rat anti-CD21/35-Fab-Atto590 7G6 | DRFZ Berlin | – | 10 µg/mouse |
| Antibody | Hamster anti-IgD-Alexa594 11.26 c | DRFZ Berlin | --- | – |
| Antibody | Rat anti-CD21/35-Alexa647 7G6 | DRFZ Berlin | – | – |
| Antibody | Rat anti-CD19-Cy5 1D3 | DRFZ Berlin | – | – |
| Antibody | Rat anti-CD40 3/23 | BD Pharmingen | Cat# 553788 | As indicated |
| Antibody | Goat anti-IgM-F(ab)$_2$ polyclonal | Southern Biotech | Cat#1023-01 | As indicated |
| Antibody | Rat anti-kappa 187.1 | DRFZ Berlin | – | As indicated |
| Recombinant DNA reagent | CpG | Tib Molbiol Berlin | Request #1668 and 1826 sequence ID: 1746437/8 | As indicated |
| Peptide, recombinant protein | Peanut-agglutinin, biotinylated | Vector Biolabs | Cat# B-1075-5 | 1:200 |
| Other | Lipopolysaccharide (LPS) from *Escherichia coli* | Sigma | Cat# L4391 | As indicated |
| Peptide, recombinant protein | NP-CGG ratio > 20 | Biosearch Technologies | Cat# N-5055C-5-BS | 10 µg/mouse |
| Peptide, recombinant protein | Streptavidin-Alexa555 | Thermo Fisher Scientific | Cat# S32355 | 1:2000 |
| Other | X-Rhod-1 | Molecular Probes | Cat# X14210 | Manufacturer's protocol |
| Software, algorithm | Python 2.7 | Python Software Foundation | | |
| Software, algorithm | MATLAB | MathWorks | | |
| Software, algorithm | Imaris | Bitplane | | |
| Software, algorithm | Phasor-based-numerical analysis of eCFP lifetime data | Own work, https://github.com/ulbrica/Phasor-FLIM.git; swh:1:rev:8ae5bfc17ec019fcc8ec7e4627442646e52cc3c5; *Ulbricht, 2021*. | | |

## Mice

Mice carrying a STOP cassette flanked by two loxP sites upstream of the region encoding for the TN-XXL biosensor (*Mank et al., 2008*) in the ROSA26 locus were obtained from F. Kirchhoff, Saarland University, Homburg. YellowCaB mice were generated by crossing those mice with the *Cd19*$^{cre/cre}$ strain (*Rickert et al., 1997*) and maintained on a C57Bl/6 background. Only YellowCaB mice heterozygous for Cre were used to avoid deletion of CD19. Mice with monoclonal NP-specific BCR (B1-8$^{hi}$:YellowCaB) were generated by crossing of YellowCaB mice with B1-8$^{hi}$ mice (*Shih et al., 2002b*). All mice were bred in the animal facility of the DRFZ. All animal experiments were approved by Landesamt für Gesundheit und Soziales, Berlin, Germany, in accordance with institutional, state, and federal guidelines.

## Cells

Primary splenocytes were isolated from whole spleens of YellowCaB mice or B1-8$^{hi}$:YellowCaB mice in 1× PBS and erythrocytes lysed. B cells were negatively isolated using the Miltenyi murine B cell isolation kit via magnetic-assisted cell sorting (MACS), leaving B cells untouched in order not to pre-stimulate them.

## Staining and flow cytometry

Single-cell suspensions were prepared and stained according to the guidelines for flow cytometry and cell sorting in immunological studies (*Cossarizza et al., 2019*). To simultaneously assess calcium influx with a dye-based method, we stained whole splenocytes or isolated B cells with the calcium-sensitive dye X-Rhod-1 (Invitrogen). X-Rhod-1 is a single-fluorophore calcium reporter molecule that enhances its fluorescence intensity upon calcium binding in a range of 0–40 µM up to 100 times at a wavelength of 600 nm. Measurements were carried out at a BD Fortessa flow cytometer. TN-XXL expression was checked assessing positive fluorescence in a 525 ± 25 nm channel after 488 nm excitation on a MACSQuant flow cytometer.

## Perfusion chamber

All in vitro experiments were carried out in Krebs–Ringer solution containing 6 mM Ca$^{2+}$ at 37°C. Cells were stimulated with anti-mouse IgM-F(ab)$_2$ (Southern Biotech), ionomycin (4 µM, Sigma), anti-CD40 antibody (BD), LPS (20 µg/ml, Sigma), or CpG (10 µg/ml, TIB Molbiol Berlin). Cell culture imaging experiments with ionomycin stimulation were performed using an open perfusion chamber system. Buffer solution was pumped through the heated chamber containing a poly-D-lysine-coated glass slide on which freshly and sterile isolated YellowCaB cells were grown for approximately 1 hr. Ionomycin was added in the flow-through buffer supply. The lag time for the volume to arrive at the imaging volume was determined for each set-up and considered for analysis of ΔR/R over time. Anti-IgM-F(ab)$_2$ antibody was given directly to cells within the open chamber in between acquisition time points. To visualize the reversibility of the sensor despite antibody still present, the experiment was performed in an open culture system without media exchange through a pump. To analyze if Yellow-CaB cells could repeatedly be stimulated, experiments were performed under continuous perfusion. Buffer flow was switched off with stimulation for several minutes and switched on again to dilute antibody out again for a second stimulation.

For analysis, regions of interest were determined based on randomly chosen single cells. Intensity density values of the raw citrine signal were divided by the intensity density values of the raw eCFP signal and related to the baseline ratio of the signals before stimulation.

## Cell transfers, immunization, and surgical procedures

B cells from spleens of YellowCaB mice were negatively isolated using the Miltenyi murine B cell isolation kit via MACS. 5 × 10$^6$ cells were transferred to a host mouse with a transgenic BCR specific for an irrelevant AG (myelin oligodendrocyte glycoprotein). When NP-specific B cells were analyzed, B cells from spleens of B1-8$^{hi}$YellowCaB mice were transferred to wildtype C57Bl/6 mice. Host mice were immunized in the right footpad with 10 µg NP-CGG in complete Freund's adjuvant 24 hr after B cell transfer. After 6–8 days p.i., FDCs were labeled with Fab-Fragment of CD21/35-Atto590 or CD21/35-Alexa647 (in-house coupling) into the right footpad. Polyclonal B cells from YellowCaB mice were stained with a red fluorescent dye (CellTracker Deep Red, Thermofisher) and

adoptively transferred. 24 hr later, the popliteal lymph node was exposed for two-photon imaging as described before (*Ulbricht et al., 2017*). Briefly, the anesthetized mouse is fixed on a microscope stage custom-made for imaging the popliteal lymph node. The mouse is shaved and incisions are made to introduce fixators that surround the spine and the femoral bone. The mouse is thus held in a planar position to the object table. The right foot is fixed by a wire allowing to increase the tension on the leg to position the lymph node parallel to the imaging set-up. A small incision is made to the popliteal area. The lymph node is exposed after freeing it from surrounding adipose tissue. A puddle around the lymph node is formed out of insulating silicon compound, then filled with NaCl and covered bubble-free with a cover slide.

For intravital application of AG, 500 µg NP-BSA were administered i.v. after acquiring 10 time steps of baseline FLIM signal (four mice). To check for BCR specificity of calcium elevation recorded after AG injections, BTK inhibitor ibrutinib was pre-injected i.v. before AG application at 3.75 mg/kg and recorded in a control group of three mice. Technically, measurements were paused for injections for about 5 min. For accurate comparison of baseline calcium levels with calcium levels after injections, we imaged the same GC before and after. Therefore, measurements are always to be treated as separate measurements and it is not possible to track individual cells before and after an injection. However, the mean calcium elevation in the presented subset of cells can be visualized. For better comparability, we have chosen to present a time course that virtually combines two consecutive measurements in the same GC into one (*Figure 5—figure supplement 1*). Up to 22 individual cell tracks were randomly chosen after gating out overlapping signals from macrophages, filtering for maximum track duration and completeness of the series of events (absolute calcium value in the center of the segmented object).

## Intravital and live cell imaging and image analysis

Imaging experiments of freshly isolated B cells were carried out using a Zeiss LSM 710 confocal microscope. Images were acquired measuring 200–600 frames with one frame/3 s frame rate while simultaneously detecting eCFP and citrine signals at an excitation wavelength of 405 nm.

For intravital two-photon ratiometric imaging, z-stacks were acquired over a time period of 30–50 min with image acquisition every 30 s. eCFP and citrine were excited at 850 nm by a fs-pulsed Ti:Sa laser, and fluorescence was detected at $466 \pm 30$ nm or $525 \pm 25$ nm, respectively. Fluorescence signals of FDCs were detected in a $593 \pm 20$ nm channel. For experiments including macrophage staining, the fluorescence data has been unmixed for a possible overlap of the TN-XXL–citrine signal with that of the injected marker to prevent false-positive colocalization analysis between the red efluor660 coupling of anti-CD169 and the green fluorescence of TN-XXL in the $525 \pm 25$ nm channel (*Rakhymzhan et al., 2017*).

For intravital FLIM experiments, eCFP fluorescence lifetime was measured with a time-correlated single-photon counting system (LaVision Biotec, Bielefeld, Germany). The fluorescence decay curve encompassed 12.4 ns (80 MHz laser repetition rate) with a time resolution of 55 ps. The pixel dwell time was $4 \times 5$ µs, allowing to detect photons from 1600 laser pulses for the fluorescence decay. The fluorescence decay, while being multiexponential, may be approximated by a bi-exponential function containing the two monoexponential decays of unquenched CFP and of FRET-quenched CFP, respectively. The phasor approach allows us to display the data prior to data interpretation graphically, that is, prior to the decision on the multiexponentiality of the CFP decay function, and was primarily used for the FRET–FLIM data evaluation. Similar to fluorescence intensity two-photon experiments, we performed time-lapse FRET–FLIM measurements and repeated the described acquisition every 30 s.

## Analysis of two-photon data

For ratiometric analysis of two-photon data, fluorescence signals were corrected for spectral overlap (the eCFP to citrine ratio in $525 \pm 25$ nm channel is 0.52/0.48) and refined by taking into account the sensitivity of photomultiplier tubes (0.37 for $466 \pm 30$ nm and 0.4 for $525 \pm 25$ nm). Ratiometric FRET for in vivo experiments was calculated accordingly as

$$FRET = \frac{1,2 \cdot ch2}{2,7 \cdot ch1 + 2,5 \cdot ch2} \cdot 100 \qquad (12)$$

Evaluation of FLIM data was performed using the phasor approach (*Digman et al., 2008*; *Leben et al., 2018*). Briefly, the fluorescence decay in each pixel of the image is Fourier-transformed at a frequency of 80 MHz and normalized, resulting into a phase vector, with the origin at (0|0) in a Cartesian system, pointing into a distinct direction within a half-circle centered at (0.5|0) and a radius of 0.5. For pure substances, vectors end directly on the half-circle, for mixtures of two on a connecting segment between the respective pure lifetimes and within a triangle, if three substances are present, and so on. The distance between several fluorescence lifetimes on the half-circle is naturally distributed logarithmically, with the longest lifetimes closer to the origin. In the case of TN-XXL, the extremes are the unquenched CFP fluorescence (2312 ps) and the eCFP fluorescence completely quenched by FRET (744 ps). The location of the measured lifetime on the connecting line can directly be translated into the amount of either eCFP state and thus to a corresponding calcium concentration in each pixel of the image.

At low signal-to-noise ratio values, the FLIM signal of the donor with a large contribution of electronic noise is shifted towards the origin of coordinates in the phasor plot, indicative for the infinite lifetime of noise. In non-fluorescent medium, we measured electronic noise and Gaussian-fitted the histograms of real and imaginary parts. The Gaussian fit of each part gives the mean (distribution center) as well as the full distribution width at half maximum, FWHM = $2\sqrt{2ln2}\sigma$, which was the same for both parts. The width of electronic noise distribution gives the radius within which we expect only noise (*Figure 4g*). In order to increase the accuracy, we excluded all data points in an area within the radius of ¾FWHM = 0.3.

### Titration of TN-XXL construct

Sensor calibration was performed using lysate of cultured, homozygous B1-8[hi]:YellowCaB plasma blasts induced from isolated B1-8[hi]:YellowCaB cells stimulated with LPS/IL-4 for 2 days. Briefly, cells were freeze-thawed in liquid nitrogen 3–4 times and treated with ultrasound for 15 min. Lysate was filtered and cell clumps separated by high-speed centrifugation. Lysis was done at two equal shares in a sufficiently small volume of calcium-free calibration buffer (Life Technologies) or calcium-saturated (39 µM CaEGTA), respectively. Calcium buffer concentrations measured were achieved by dilution of 39 µM-buffered cell lysate with 0 µM-buffered cell lysate. Sample concentrations were loaded into glass microscope slides with recess, covered, and their fluorescence was measured at the two-photon microscope in a time-resolved manner. Focus was adjusted to a z-position with maximal photon counting numbers, as ensured by acquisition of proper decay curves. In time domain, the eCFP mean fluorescence lifetimes $\tau$ at various free calcium concentrations were determined by approximating the corresponding fluorescence decay curves *F(t)* acquired with our TCSPC-based FLIM detector to a monoexponential function containing background $y_0$: $F(t) = y_0 + A \cdot e^{-\frac{t}{\tau}}$. Fitting was performed iteratively using a Levenberg–Marquardt gradient algorithm (*Rinnenthal et al., 2013*).

### Statistical information

Time-dependent FRET curve analysis shows representative graphs for the number of analyzed cells and independent experiments given. For multiple curve analysis, mean is shown and SD indicated in each data point. For column analysis, one-way ANOVA with Bonferroni multiple comparison test was applied with a confidence Interval of 95%.

### Data availability

All raw data and analyzed data shown here are stored on institutional servers. Imaging source data and raw Excel files have been deposited at https://datadryad.org under DOI: 10.5061/dryad.cc2fqz63d.

## Acknowledgements

We thank Patrick Thiemann, Vivien Theissig, and Manuela Ohde for animal caretaking. We thank Robert Günther for excellent surgical assistance and Peggy Mex for cell isolations and stainings. We thank Ralf Uecker for microscope facility services. We further thank Mathis Richter, who provided support with the SNR-based quality check of imaging data and their evaluation. This study has been

supported by the Deutsche Forschungsgemeinschaft (DFG) TRR130, project 17 (to AEH and HR) and C01 (to AEH and RAN), and DFG SFB 1444, project 14 (to AEH and RAN).

## Additional information

### Competing interests

Frank Kirchhoff: Reviewing editor, *eLife*. The other authors declare that no competing interests exist.

### Funding

| Funder | Grant reference number | Author |
|---|---|---|
| Deutsche Forschungsgemeinschaft | TRR130 P17 | Helena Radbruch<br>Anja E Hauser |
| Deutsche Forschungsgemeinschaft | TRR130 C01 | Raluca A Niesner<br>Anja E Hauser |
| Deutsche Forschungsgemeinschaft | TRR130 P04 | Lars Nitschke |
| Deutsche Forschungsgemeinschaft | SFB1444 P14 | Raluca A Niesner<br>Anja E Hauser |

The funders had no role in study design, data collection and interpretation, or the decision to submit the work for publication.

### Author contributions

Carolin Ulbricht, Conceptualization, Data curation, Formal analysis, Validation, Visualization, Methodology, Writing - original draft; Ruth Leben, Data curation, Software, Formal analysis, Validation, Investigation, Visualization, Methodology, Writing - review and editing; Asylkhan Rakhymzhan, Data curation, Software, Formal analysis, Investigation, Methodology; Frank Kirchhoff, Resources; Lars Nitschke, Resources, Methodology; Helena Radbruch, Supervision, Funding acquisition, Writing - original draft; Raluca A Niesner, Conceptualization, Resources, Data curation, Software, Formal analysis, Supervision, Funding acquisition, Validation, Investigation, Visualization, Methodology, Project administration, Writing - review and editing; Anja E Hauser, Conceptualization, Resources, Formal analysis, Supervision, Funding acquisition, Validation, Investigation, Methodology, Project administration, Writing - review and editing

### Author ORCIDs

Carolin Ulbricht (iD) https://orcid.org/0000-0003-2983-6242
Asylkhan Rakhymzhan (iD) https://orcid.org/0000-0003-3152-1557
Anja E Hauser (iD) https://orcid.org/0000-0002-7725-9526

### Ethics

Animal experimentation: The study was approved by the Berlin Landesamt für Gesundheit und Soziales under the registration # G00158/16. All surgeries and experimental procedures were conducted following the principle of minimization of suffering and 3R means were used where possible.

### Decision letter and Author response

Decision letter https://doi.org/10.7554/eLife.56020.sa1
Author response https://doi.org/10.7554/eLife.56020.sa2

## Additional files

### Supplementary files

- Source code 1. Annotated Python-based code for phasor analysis.
- Transparent reporting form

## Data availability

Source data for flow cytometric Analysis, in vitro confocal imaging, ratiometric in vivo Imaging and fluorescence lifetime in vivo Imaging are deposited at Dryad Digital Repository 10.5061/dryad. cc2fqz63d. Analyzed absolute calcium concentration for all cells measured out of 5 experiments have also been deposited there. Source code for phasor based analysis of fluorescence lifetime data has been provided with full submission upload and is available to the public via github (https://github.com/ulbrica/Phasor-FLIM; https://archive.softwareheritage.org/swh:1:rev:8ae5bfc17ec019fcc8ec7e4627442646e52cc3c5).

The following dataset was generated:

| Author(s) | Year | Dataset title | Dataset URL | Database and Identifier |
|---|---|---|---|---|
| Ulbricht C, Leben R, Rakhymzhan A, Kirchhoff F, Nitschke L, Radbruch H, Niesner RA, Hauser AE | 2021 | Intravital quantification reveals dynamic calcium concentration changes across B cell differentiation stages | https://doi.org/10.5061/dryad.cc2fqz63d | Dryad Digital Repository, 10.5061/dryad.cc2fqz63d |

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
