## [Decision Letter]

**Acceptance summary:**

You have generated a new FLIM based reporter to follow cytoplasmic Calcium homeostasis in B cells undergoing germinal center responses and differentiation into plasma cells. You have also introduced a new analytical framework based on the phasor method, which provides a new tool for study of Calcium signals in vivo. You have also drawn attention to the heterogeneity of in vivo cytoplasmic Calcium in B cells and plasma blasts. A limitation of the study is that direct comparison to other established methods were not performed, which means that interpretation of the heterogeneity is not straight forward for others in the field and teams considering the new method will need to undertake testing to determine if this method outperforms others for a particular application. Nonetheless, the new tools developed here and initial findings should motivate further efforts to understand the role of cytoplasmic Calcium heterogeneity in, for example, vaccination responses, which we all appreciate.

**Decision letter after peer review:**

Thank you for sending your article entitled "Intravital quantification of cytoplasmic B cell calcium reveals dynamic signaling across B cell differentiation stages" for peer review at *eLife*. Your article is being evaluated by 2 peer reviewers, one of whom is a member of our Board of Reviewing Editors, and the evaluation is being overseen by Satyajit Rath as the Senior Editor.

There are technical concerns about your calibration that may be addressed without new experiments. There is a larger concern regarding the nature of the Calcium signals and evidence that these are dependent upon the BCR. It’s not felt that the observation of Ca^2+^ fluctuations along in GC B cells or plasma blasts is unexpected or sufficient. Can the authors demonstrate in either the of these signals is actually dependent upon the BCR? Perhaps this could be done in the setting of B1.8 setting with non-toxic versions of the hapten, that blocking the BCR can prevent Calcium signaling in either or both compartments?

Reviewer #1:

The authors establish a mouse line expressing a Fret based Calcium sensor in B cells and they perform life-time imaging to determine the cytoplasmic Ca^2+^ concentration in B cells activated through their antigen receptor in vitro or as part of a germinal center reaction. A strength of the approach is that the lifetime imaging gives a result that is independent of depth assuming the signal to noise ratio is adequate to acquire robust data. Using this approach they find that germinal center B cells and plasma blasts in the medullary cords both have elevated Ca^2+^. The non-antigen specific control cells are presumably naïve T cells, which are found everywhere in the lymph nodes and thus measurements could be made in GC or MC. The weakness of using these cells as controls for antigen recognition is that they are follicular B cells, not GC B cells or plasma blasts. So not only is their antigen receptor different, but chemokine receptors and other receptors for environmental signals will be different in these cells. This seems to mainly be written as a resource papers to illustrate the methods, establish that Ca^2+^ fluctuations can be measured, but is limited in the extent to which these differences can be attributed to antigen in vivo as antigen recognition is linked to both acute activation processes and differentiation into different B cell types.

1. The interpretation of any of the data as antigen specific is difficult as the antigen specific cells have distinct differentiation compared to the control polyclonal cells. In order to assess this, a way to block the antigen receptors provide a bolus of antigen to directly observe antigen dependent changes in Ca^2+^ in vivo could be useful.

2. The observations that the plasma blasts in the MC have high Ca^2+^ fluctuations is interesting, but may be related to chemokine dependent interactions with macrophages or closely associated stromal cells (see: Fooksman DR, Schwickert TA, Victora GD, Dustin ML, Nussenzweig MC, Skokos D. Development and migration of plasma cells in the mouse lymph node. Immunity. 2010;33(1):118-27 and Huang HY, Rivas-Caicedo A, Renevey F, Cannelle H, Peranzoni E, Scarpellino L, Hardie DL, Pommier A, Schaeuble K, Favre S, Vogt TK, Arenzana-Seisdedos F, Schneider P, Buckley CD, Donnadieu E, Luther SA. Identification of a new subset of lymph node stromal cells involved in regulating plasma cell homeostasis. Proc Natl Acad Sci U S A. 2018;115(29):E6826-E35.). This Calcium signaling is unlikely to be antigen dependent so they should discuss alternatives and if possible devise an experiment to directly address this issues, at least for antigen. Can they influence the signaling status of the plasma blasts with antigen at all?

Reviewer #2:

Ulbricht et al., employed a FRET-based calcium (Ca^2+^) sensor YellowCaB to study Ca^2+^ signaling in B cells. in vitro experiments using fluorescent intensities of eCFP and citrine demonstrate the ability of YellowCaB to report repeated BCR stimulation. The authors show examples of Ca^2+^ transients in B cells contacting FDCs and other B cells during intravital imaging. Fluorescence lifetime analysis revealed multiple lifetimes for FRET-donor eCFP. Based on the Kd of 453 nM for TN-XXL and phasor plot analysis, the authors calculate the absolute Ca^2+^ concentration in B cells. B cells display heterogeneity in calcium signals. Surprisingly, extrafollicular B cells displayed higher Ca^2+^ levels that correlated with the duration of contact with subcapsular sinus macrophage contact. However, there are important concerns.

1. Calibration. Characterizing Ca^2+^ signaling in B cells during immune responses will likely be important for understanding affinity maturation. However, FLIM-based analysis needs further controls to validate the claim of absolute Ca^2+^ concentrations reported in this study. Geiger et, al (Biophysical Journal, May 2012) cautioned for FRET-based biosensors such as TN-XXL, that calibration should be performed on the same microscope set up and the same conditions as used for imaging. It is unclear from the Methods how the lifetime of eCFP 2300ps (unquenched) vs 700ps (quenched) was determined, e.g., at what pH and temperature? Ideally, FLIM probes should have monoexponential decay curves, what is the lifetime of eCFP alone expressed in B cells?

2. Expression of TN-XXL sensor in one allele (TN-XXL^+/-^) vs homozygous (TN-XXL^+/+^) animal should be different in intensity but not in the percent of B cells positive for TN-XXL. This should be clarified.

3. Moreover, it appears that there were no new insights resulting from the calibrated Ca^2+^ concentrations. It is not unexpected that there is heterogeneity in the calcium responses.

[Editors' note: further revisions were suggested prior to acceptance, as described below.]

Thank you for submitting your article "Intravital quantification reveals dynamic calcium concentration changes across B cell differentiation stages" for consideration by *eLife*. Your article has been reviewed by 3 peer reviewers, including Michael L Dustin as the Reviewing Editor and Reviewer #1, and the evaluation has been overseen by Satyajit Rath as the Senior Editor.

The reviewers have discussed the reviews with one another and the Reviewing Editor has drafted this decision to help you prepare a revised submission.

We would like to draw your attention to changes in our revision policy that we have made in response to COVID-19 (https://elifesciences.org/articles/57162). Specifically, we are asking editors to accept without delay manuscripts, like yours, that they judge can stand as *eLife* papers without additional data, even if they feel that they would make the manuscript stronger. Thus the revisions requested below only address the validity of claims based on the data shown, clarity and presentation.

The reviewers, two of whom are new due to the unavailability of one of the original reviewers, appreciate your efforts to address the original concerns. The consensus is that you have succeeded in calibrating the lifetime measurements, with one caveat that the use of the lysate may not fully recapitulate the conditions in a live cell, but since it's difficult to clamp Calcium in a live cell your solution is acceptable. The experiments addressing the role of antigen in germinal centre B cell activation are considered helpful as a sensitivity test, but there is a consensus that your new text over-interprets the data presented. You have established BCR triggered Calcium flux in vitro (original data) and in vivo (new data with antigen injection), but this capability doesn't appear to help understand the heterogeneity of B cell's Calcium levels in the germinal centre or medullary cords. The reviewers can see a positive path and are happy to see another revision if you can respond to the new concerns, which largely relate to refining presentation of the new information introduced in your first revision.

The calibration of the lifetime data is appreciated, but some discussion should be given to limitations of the approach, which are unavoidable, but requires caution.

The NP-BSA induced Calcium elevation in vivo extends results with anti-Ig from Figure 2 and supports ability to detect BCR dependent Calcium ion elevation in vivo. NP-BSA can trigger apoptosis (Nossal, PMID 7753199) and that interactions with myeloid cells in the MC seems to limit antibody secreting cells numbers (Fooksman, PMID 24376270) such that signals leading to apoptosis may also be contributing to the Calcium signals observed in the system. This could be incorporated into the discussion of different explanations for the high Calcium subsets. It would also make sense to cite early work from Cahalan that Calcium elevation can lead to arrest of cell motility (Negulescu et al., PMID: 8630728). But it's important to point out that demonstrating that you can detect effects of BCR engagement by injecting polyvalent antigen is not the same as showing that Calcium elevation observed in germinal centre and medullary cords in the system set up by earlier immunisation is due to antigen recognition. In fact, the apparent lack of effect of the BTK inhibitor doesn't support a role for BCR signalling in the heterogeneity of Calcium.

The new data in Figure 5a is appreciated as it attempts to address the role of BCR signalling using a BTK inhibitor and injecting NP-BSA as a BCR ligand. The description of the experiments suggests that you have performed a very demanding time-course with a baseline, BTK inhibitor injection and then NP-BSA injection. Could this data be shown- similar to Figure 4g, but with breaks where the injections were performed. There are questions about how long it takes these agents to get to the lymph nodes after IV injection. The inhibitor may be fine, but I'm not sure the NP-BSA will get to the germinal centre in a lymph node by this route as it will need to leak through the vasculature into the peripheral tissues and then drain to the LN. Do the authors know of data on BSA pharmacodynamics that would suggest it gets into the germinal centre in minutes after IV injection? I know of data for peptides getting into the T cell zone rapidly, but not for an intact protein that is maintained in the blood by FcRn. I would think that marginal zone of spleen would be the main point of entry for IV protein into secondary lymphoid tissues.

As mentioned above, you need to think carefully about reasonable interpretations of these data and revise the discussion regarding possible sources of Calcium heterogeneity.

[Editors' note: further revisions were suggested prior to acceptance, as described below.]

Thank you for sending your article entitled "Intravital quantification reveals dynamic calcium concentration changes across B cell differentiation stages" for peer review at *eLife*. Your article is being evaluated by 1 peer reviewers, and the evaluation is being overseen by a Reviewing Editor and Satyajit Rath as the Senior Editor.

Your last revision didn't appear to address the reviewers concerns, but the Reviewing Editor asked for another opinion in an attempt to pinpoint and articulate the key issues related to calibration. The opinion below agrees that the revision didn't address reviewer concerns, but suggests a different approach to the Phasor calibration that may help resolve this issue, although it would still require some adaptation from the original use with a Calcium indicator to the current use with a Calcium biosensor. The Reviewing Editor hopes that this will be helpful in focusing a final revision that can be resent to all active reviewers.

"I found this article quite confusing in regard to the calcium calibration and the proper citation of the phasor method. The authors fail to cite the paper using the phasor approach for the calcium calibration "Celli A, Sánchez SA, Behne MJ, Hazlett TL, Gratton E, Mauro TM. The epidermal Ca(2+) gradient: measurement using the phasor representation of fluorescent lifetime imaging. Biophys J. 2010; 98(5): 911-921. PMCID: PMC2830439. This paper addresses the issue of the calibration of indicators, not biosensors. It uses a different method based on the position of the data on the phasor plot rather than attempting a calibration that cannot be done due to the diversity of the calibration solutions with respect to the environment where the biosensor is located, which is a major point of the previous comments.

The only phasor plots in this paper are in figure 4, where the phasor of the biosensor is located but in my opinion the calibration method suggested in the paper by Celli et al. is not applied. This is important for the overall interpretation of the results and the conclusion of this article."

---

## [Author Response]

There are technical concerns about your calibration that may be addressed without new experiments. There is a larger concern regarding the nature of the Calcium signals and evidence that these are dependent upon the BCR. It’s not felt that the observation of Ca^2+^ fluctuations along in GC B cells or plasma blasts is unexpected or sufficient. Can the authors demonstrate in either the of these signals is actually dependent upon the BCR?

We thank the reviewing editor for the honest interest in our manuscript and for the helpful suggestions, which helped us improve our research. We agree that providing a proper calibration of the TN-XXL Ca^2+^ sensor is indispensable for the reliability of the data. Therefore, we performed a new set of FLIM-measured calibration experiments, and also included already existing, mono- and multiexponentially analyzed lifetime data. Concerns regarding the nature of intra- and extrafollicular Ca^2+^ signaling events were addressed in vitro (by FLIM of stimulated LPS-induced plasma blast cultures) as well as in vivo (by injection of antigen, as suggested by the reviewers, and by adding a BCR-inhibitor while intravital imaging was performed).

Perhaps this could be done in the setting of B1.8 setting with non-toxic versions of the hapten, that blocking the BCR can prevent Calcium signaling in either or both compartments?

We followed the suggestion of measuring potential signaling events in a B1-8 (high affinity BCR for NP) setting after application of antigen. To do so, we performed intravital imaging experiments at day 7 and 8 after immunization within the popliteal lymph node, as described. This time, after acquiring basal calcium concentrations in GC B cells (identified by proximity to FDC staining with anti-CD21/35), mice received an intravenous injection of 100µl NP(7)-BSA solution, representing 500µg of antigenconjugate per animal. We were able to observe four GCs in three mice in total. In every case, injection of NP-BSA significantly increased overall Ca^2+^ concentrations within the GC B cell population, with a mean calcium elevation of ca. 150nM. While, due to the proposed degree of heterogeneity within the GC, this is likely not affecting all GC B cells, the single-cell data suggest a concentration-shift in a proportion of GC B cells that is sufficient to indicate a statistically significant elevation in the overall cytoplasmic calcium level. See Author response image 1.

**Author response image 1. respfig1:** A: intravital imaging of GC B cells before and after i. v. application of NP-BSA B: intravital imaging of GC B cells before and after i.v. application of ibtutinib, followed by NP-BSA injection. C: Cumulated data of revised intravital imaging experiments.

To address BCR-specificity of these signals further, we pre-injected ibrutinib, which is used to efficiently block BCR signaling by preventing the phosphorylation of molecules downstream of BTK, a key adaptor enzyme of the BCR signaling cascade (Herman, Blood, 2011). in vivo treatment of mice with ibrutinib (110µg/kg i.v.) before injection of NP-BSA was able to reduce Ca^2+^ increase, though not to completely abrogate it. We conclude that, at least in part, the calcium fluctuations observed in our in vivo data are BCR specific, but that BCR-mediated signaling is not the only source of cytoplasmic Ca^2+^ heterogeneity in the GC. We want to point out here that we did not aim on providing a tool for measuring BCR-specific signaling events in the first place, but rather for signaling events mediated by Ca^2+^ as a whole. We believe that further experiments will disseminate the nature of cytoplasmic calcium changes even further, some of them already performed in our lab (our preliminary data indicate a role for metabolic activity thresholds in Ca^2+^ homeostasis). We emphasized these points within the text.

Reviewer #1:The authors establish a mouse line expressing a Fret based Calcium sensor in B cells and they perform life-time imaging to determine the cytoplasmic Ca^2+^ concentration in B cells activated through their antigen receptor in vitro or as part of a germinal center reaction. A strength of the approach is that the lifetime imaging gives a result that is independent of depth assuming the signal to noise ratio is adequate to acquire robust data. Using this approach they find that germinal center B cells and plasma blasts in the medullary cords both have elevated Ca^2+^. The non-antigen specific control cells are presumably naïve T cells, which are found everywhere in the lymph nodes and thus measurements could be made in GC or MC. The weakness of using these cells as controls for antigen recognition is that they are follicular B cells, not GC B cells or plasma blasts. So not only is their antigen receptor different, but chemokine receptors and other receptors for environmental signals will be different in these cells. This seems to mainly be written as a resource papers to illustrate the methods, establish that Ca^2+^ fluctuations can be measured, but is limited in the extent to which these differences can be attributed to antigen in vivo as antigen recognition is linked to both acute activation processes and differentiation into different B cell types.

We thank reviewer #1 for these comments and want to point out that indeed the non-antigen specific cells are naive, that is primary, most likely follicular B cells with polyclonal BCRs, carrying the TN-XXL construct. These cells were isolated via CD19 MACS from YellowCaB mice and adoptively transferred into wild-type recipients one day prior to imaging. One obstacle with these cells is the fact that TN-XXL fluorescence within these cells is faint due to their low amount of cytoplasm, and therefore only suboptimal signal-to-noise conditions can be met in most cases. Therefore, we decided to exclude all cells from the analysis with SNR<1 (for AG-specific cells SNR<2). It is true that these polyclonal cells also might possess differences in receptor composition. However, rather than to prove antigen-specificity of the signaling by use of these cells as control, we wanted to characterize these differences by showing the different degrees of heterogeneity among polyclonal cells in contrast to AG-specific ones. In the AG-specific subset (including B cells and plasma blasts), we see a distinct population arising that is showing elevated Ca^2+^ concentrations. We hypothesize that a certain Ca^2+^ level is associated with selection and B cell survival, which is likely the reason why naive, polyclonal B cells have relatively and uniformly low Ca^2+^ concentrations.

1. The interpretation of any of the data as antigen specific is difficult as the antigen specific cells have distinct differentiation compared to the control polyclonal cells. In order to assess this, a way to block the antigen receptors provide a bolus of antigen to directly observe antigen dependent changes in Ca^2+^ in vivo could be useful.

We thank the reviewer for this comment and feel that antigen-stimulation experiments in our system are by right an issue of strong interest. In accordance with suggestions of reviewer #1 and the reviewing editor, new in vivo data have been generated, showing i.v. injection of NP is able to increase Ca^2+^ in GC B cells. In addition, we performed experiments using the BCR inhibitor ibrutinib, which demonstrated that Btk-inhibition is able to restrict this cytoplasmic Ca^2+^ increase (see answer to the reviewing editor above).

2. The observations that the plasma blasts in the MC have high Ca^2+^ fluctuations is interesting, but may be related to chemokine dependent interactions with macrophages or closely associated stromal cells (see: Fooksman DR, Schwickert TA, Victora GD, Dustin ML, Nussenzweig MC, Skokos D. Development and migration of plasma cells in the mouse lymph node. Immunity. 2010;33(1):118-27 and Huang HY, Rivas-Caicedo A, Renevey F, Cannelle H, Peranzoni E, Scarpellino L, Hardie DL, Pommier A, Schaeuble K, Favre S, Vogt TK, Arenzana-Seisdedos F, Schneider P, Buckley CD, Donnadieu E, Luther SA. Identification of a new subset of lymph node stromal cells involved in regulating plasma cell homeostasis. Proc Natl Acad Sci U S A. 2018;115(29):E6826-E35.).

We agree with the reviewer’s notion that chemokine-induced Ca^2+^ mobilizations within lymphocytes are common and their impact should be taken into account in this system. During establishment of our ratiometric system, we already tried to address the question whether such Ca^2+^ changes are being detected, and these data were included in the supplementary material of the original manuscript (Figure S2c). In our hands, the ratiometric imaging approach was not sensitive enough to detect calcium mobilization in primary, polyclonal YellowCaB cells. However, this of course does not mean that the calcium fluctuations measured in our in vivo set up might not as well be a result of chemokine stimulation, at least partially. Furthermore, as pointed out by reviewer #1, AG-specific B cells and plasma blasts might have different receptor compositions than follicular B cells. Also, FLIM measurements might be more sensitive than ratiometric imaging. Therefore, and in order to investigate this issue further, we decided to repeat CXCL12-stimulated Ca^2+^-measurements in cultured B cells (LPS/IL-4 for 2 days) and tested if we could detect Ca^2+^ signaling with FLIM after direct CXCL12 stimulation. We can conclude that Ca^2+^ changes in YellowCaB plasmablasts are measurable with FLIM after chemokine stimulation, and that chemokine-induced Ca^2+^ fluctuations are thus likely to contribute to Ca^2+^ heterogeneity seen in our in vivo data, at least at sites with high local concentration. Figure S2 has been updated accordingly.

This Calcium signaling is unlikely to be antigen dependent so they should discuss alternatives and if possible devise an experiment to directly address this issues, at least for antigen. Can they influence the signaling status of the plasma blasts with antigen at all?

Plasmablasts and even plasma cells of the IgA and IgM isotype can carry residual BCR activity, as recently reported by the lab of L. Sollid (Spencer and Sollid, 2016). In our experiments, we could demonstrate Ca^2+^ elevations in B1-8-YellowCaB plasma blasts by FLIM after addition of the antigen (NP) (Author response image 2).

**Author response image 2. respfig2:** CXCL12 stimulation of LPS-induced B1-8 PB, B: NP stim of LPS-induced B1-8 PB.

We want to point out again that we did not aim on establishing the TN-XXL^+^ YellowCaB system as a readout for BCR-specific activation; rather, we wanted to provide a tool for measuring Ca^2+^ as ubiquitous, universal cellular messenger. In addition to alterations in the text, we have also changed the title in order to emphasize this point. The fact that we can measure absolute, that is, discrete concentrations of Ca^2+^ might also make it a useful tool to distinguish different Ca^2+^ signaling pathways, as each of these processes may be acting in a quantum-mediated manner that relies upon defined thresholds, a hypothesis that we are discussing in the revised version of the paper. However, to quantitatively dissect to what extent these differential signaling patterns affect Ca^2+^ levels, would go beyond the scope of our work at present.

Reviewer #2:Ulbricht et al., employed a FRET-based calcium (Ca^2+^) sensor YellowCaB to study Ca^2+^ signaling in B cells. in vitro experiments using fluorescent intensities of eCFP and citrine demonstrate the ability of YellowCaB to report repeated BCR stimulation. The authors show examples of Ca^2+^ transients in B cells contacting FDCs and other B cells during intravital imaging. Fluorescence lifetime analysis revealed multiple lifetimes for FRET-donor eCFP. Based on the Kd of 453 nM for TN-XXL and phasor plot analysis, the authors calculate the absolute Ca^2+^ concentration in B cells. B cells display heterogeneity in calcium signals. Surprisingly, extrafollicular B cells displayed higher Ca^2+^ levels that correlated with the duration of contact with subcapsular sinus macrophage contact. However, there are important concerns.1. Calibration. Characterizing Ca^2+^ signaling in B cells during immune responses will likely be important for understanding affinity maturation. However, FLIM-based analysis needs further controls to validate the claim of absolute Ca^2+^ concentrations reported in this study. Geiger et, al (Biophysical Journal, May 2012) cautioned for FRET-based biosensors such as TN-XXL, that calibration should be performed on the same microscope set up and the same conditions as used for imaging. It is unclear from the Methods how the lifetime of eCFP 2300ps (unquenched) vs 700ps (quenched) was determined, e.g., at what pH and temperature? Ideally, FLIM probes should have monoexponential decay curves, what is the lifetime of eCFP alone expressed in B cells?

We share the opinion of the reviewer that a thorough characterization of the donor (CFP) fluorescence decay in the TNXXL construct, without any quenching and under FRET-quenching, is crucial for calculating absolute Calcium values from our FRET-FLIM data. We thank the reviewer for pointing out that in the present version of the manuscript the description of the calibration is not complete and might lead to misinterpretations. In the revised version of the manuscript, we added new data and text to clarify this issue.

As suggested by the reviewer, we included FLIM data we previously acquired in spleen tissue of mice ubiquitously expressing CFP (Author response image 3). Under two-photon excitation at 850 nm and detection with our TCSPC at 460±30 nm, we could validate its fluorescence lifetime value of 2300 ps. A corresponding fluorescence decay curve averaged over a 300x300 µm² region in spleen tissue (256 x 256 pixel) was included in the manuscript, together with exemplary close-up images of CFP-expressing splenocytes (revised Figure 4). Fitting the measured CFP fluorescence decay with a mono-exponential function with background: f(t)=yo+A•etτ led to a fluorescence lifetime τ of 2303±54 ps. Approximation with bi-exponential or tri-exponential functions did not reveal additional fluorescence lifetimes, but replicated the result of the mono-exponential fit (2303±54 ps). Thus, we conclude that the approximation of the CFP fluorescence decay with a mono-exponential curve is valid. We decided to not repeat this experiment specifically in B cells, as we do not expect the lifetime to majorly differ between lymphocyte subsets, and since we aim to pursue 3R guidelines, therefore reducing the use of laboratory animals used in experiments to a minimum.

As our FRET-FLIM calibration of the TN L15 construct (Rinnenthal et al., PLoS One, 2013) – a similar construct to TN XXL– was in very good agreement with the previously published calibration curve (Heim N., Griesbeck O., J. Biol. Chem., 2004), and since our CFP fluorescence lifetime measurements in B cells were in the same range as those published by Geiger et al. (Biophys J, 2012), we considered that using the published FRET-FLIM calibration curve for TNXXL for our data is feasible. However, to exclude any possible artifacts and to validate the published calibration curve using our own experimental setup, we performed titration FRET-FLIM experiments using lysates of B cells expressing TNXXL at various, well-defined Ca^2+^ concentrations. Exemplary fluorescence decay curves of CFP in the TNXXL construct from B cell lysates at 0 nM, 602 nM and 39 µM free Ca^2+^ have been added to the revised Figure 4, showing donor quenching with increasing Ca^2+^ concentration. By fitting the CFP decay curves acquired at 0 nM, 65 nM, 150 nM, 351 nM, 602 nM, 1.35 µM and 39 µM free Ca^2+^, we generated the corresponding titration curve of the TNXXL construct (results of three independent experiments). We approximated this calibration curve with the sigmoid function: τ (lg[Ca])=τmin+tmax−tmin1+10(lg[Kd]−lg[Ca])•Hillslope (revised Figure 4) and determined the parameters: K_d_ = 475±46 nM (lg[K_d_] = -6.32±0.04) and Hill slope = 1.43±0.17. Both parameters are in very good agreement with the previously published results of Geiger et al., as differences fall within the error margins. Consequently, the resulting 4.5% difference in absolute calcium concentrations lies below our measurement accuracy. Also, the values of τ _max_ = 2322±42 ps and τ _min_ = 769±70 ps confirm the previously used lifetime values (unquenched: 2300 ps and FRET-quenched: 700 ps), and do not lead to any measurable differences in absolute calcium concentration.

Following the reviewer’s suggestion, we emphasized in the revised version of the manuscript that the fluorescence lifetime of fluorophores depends on the refractive index (Strickler and Berg, 1962, J. Chem. Phys. 37:814) and may be influenced by various other factors such as pH, temperature, or ionic strength. All our intracellular measurements were performed under typical intracellular conditions, i.e. pH 7.2-7.4, 37°C temperature, at the ionic strength of cytosol and refractive index of cytosol which is comparable to that of water (1.33). The extracellular measurements were performed under similar conditions as the intracellular measurements, except for the fact that they have been performed at room temperature. While Laine et al., PLoS ONE, 2012 reported a slight decrease of Cerulean fluorescence lifetime in the TN L15 construct in the range 20°C – 50°C, we did not observe such a trend as we measured the same CFP fluorescence lifetime both in spleen tissue at 37°C and in 0nM Ca^2+^ free buffered solutions of TN XXL, at room temperature.

Once again, we agree with the reviewer that the experimental setup may influence the results of FRET measurements. This holds true especially for ratiometric FRET, for which measuring both the fluorescence signal of the donor and of the acceptor is necessary. As detector sensitivity may vary between spectral channels and different photobleaching and scattering properties of donor and acceptor fluorophores are expected, a thorough calibration is needed not only for each experimental setup, but also for each sample type. In contrast, FRET-FLIM measures only the time-resolved fluorescence of the donor and is neither affected by the different photobleaching of donor and acceptor molecules, nor by the different scattering properties of their fluorescence. This is especially relevant in deep tissue and makes FLIM the method of choice for standardized, generally valid information on intracellular calcium concentrations, as previously shown by us in neurons (Radbruch et al., 2015, IJMS). Artifacts in fluorescence lifetime may however appear, but are mainly related to noise contribution, which – being an undamped oscillation – leads to accuracy loss. We better emphasized in the revised version of the manuscript, how we accounted for background noise impact in our phasor evaluation. We measured all fluorescence decays using TCSPC with high temporal resolution, i.e. time bin 55 ps, electronic jitter < 10 ps, and instrument response function ≈ 200 ps. Under these conditions, the fluorescence decay of unquenched CFP is best approximated by a mono-exponential function. The fluorescence decay of CFP in the TNXXL construct in the presence of Ca^2+^ is more complex, due to the fact that it represents the average over all CFP (donor in TNXXL) molecules dwelling within the observation volume. Each Tnroponin C molecule in the TNXXL construct has four different Ca^2+^-binding sites – two high-affinity and two-low-affinity binding sites, leading to different quenching levels of CFP. Accounting for the Ca^2+^-binding heterogeneity among the TN XXL molecules within the observation volume, the measured CFP fluorescence decay will be multi-exponential (at least fourexponential) and, as such, its numerical approximation with a correct fitting curve is challenging. By simplifying the situation and fitting the decay with a mono-exponential function, mean CFP fluorescence lifetimes which lie between 2300 ps (unquenched CFP) and 700 ps (completely FRET-quenched CFP) are expected.

**Author response image 3. respfig3:** A: exemplary monoexponential decays for Ca^2+^ free (black), Ca^2+^ saturated (blue) and 602nM Ca^2+^ medium (red), and calibration curve for lysate of YellowCaB cells. Measured concentrations (3 replicates): 0nM, 100nM, 150nM, 351nM, 602nM, 1,35μM, 39μM. B: unquenched fluorescence lifetime of eCFP was validated to be 2303+/- 53 ps in splenic tissue of mice with ubiquitous GFP expression. C: exemplary decays for plasma blasts with high (tau = 703 ps, red arrow) and lower (tau = 1937ps, white arrow) Ca^2+^ concentration.

In order to exclude artifacts introduced by model-based approximation algorithms, we decided to use decay visualization from the entire image, without any previous knowledge of the system (Digman M.A. et al., Biophys J., 2008). First when interpreting the evaluated data, assumptions regarding the system (especially about its molecular heterogeneity) need to be done. To validate the performance of the phasor approach in our in vivo experiments, we fitted fluorescence decay curves of plasmablasts expressing TNXXL using the Levenberg-Marquardt gradient approach to mono-exponential curves with background. The resulting fluorescence lifetimes were in good agreement with the mean fluorescence lifetime calculated using the phasor approach. We added to the revised Figure 4 representative decay curves of plasmablasts as well as their mono-exponential fits. The corresponding fluorescence lifetimes amount to 1937±49 ps and 703±56 ps. The long fluorescence lifetime indicates little FRET-quenching and, thus, a low cytosolic calcium concentration around 150 nM, however no unquenched CFP (2300 ps) indicating calcium-free medium. The short fluorescence lifetime corresponds to fully FRET-quenched CFP indicating a high calcium concentration in cytosol, above 1 µM. Additionally, these results confirm under in vivo conditions the determined τ _max_ and τ _min_ of CFP in the TNXXL construct, used for the calibration. The bi-exponential approximation of the exemplary decay curves led to τ_1_ = 563±135 ps and τ_2_ = 2335±180 ps (mean τ 1937±49 ps) and τ_1_ = 703±56 ps and τ_2_ > 15000 ps representing background noise (mean τ 703±56 ps).

2. Expression of TN-XXL sensor in one allele (TN-XXL^+/-^) vs homozygous (TN-XXL^+/+^) animal should be different in intensity but not in the percent of B cells positive for TN-XXL. This should be clarified.

We agree with the reviewer that the description on TNXXL sensor expression, as presented in the previous version of the manuscript, is inconclusive. We have revised the data and added some more recently generated data sets, where we paid particular attention in order to handle all cells uniformly careful (Author response image 4). We find that there is no evidence of differential expression between the genotypes. We explain the discrepancies of the original data with the fact that gating on YFP-Intensity alone is insufficient to draw conclusions about expression. This is because YFP Intensity is also a result of the amount of Ca^2+^ ions present within the cells, and not only of the amount of TNXXL-molecules. Within a heterogeneous population of splenic B cells, YFP-intensity comprises a spectrum rather than a uniquely assignable feature. Thus, overlaying effects like pre-activation of a proportion of cells create the impression that there are more cells in the YFP+ gate, which they can be attributed to because of their then higher YFP-intensity. On the other hand, cells not being part of this gate because of lower intensity do not equal cells that have no TNXXL expression. However, because eCFP is unquenched in these individuals, their YFP-intensity will be too low for detection in the YFP+ gate.

**Author response image 4. respfig4:** 

3. Moreover, it appears that there were no new insights resulting from the calibrated Ca^2+^ concentrations. It is not unexpected that there is heterogeneity in the calcium responses.

The comment made us realize that we might have not spent enough effort to clarify the points of our findings and we thank the reviewer for that. There are two major advances we wanted to highlight with the manuscript. The first points we would like to make are methodological. While it is true that optical Ca^2+^ measurements have a certain tradition, we are the first

– to measure absolute Ca^2+^ amounts, not only relative changes in B cells and plasma cells

– to achieve Ca^2+^ quantification in vivo, and during a process that is central to immunity

– to establish the mathematical workflow based upon phasor analysis, that is simplifying the analysis of multi-component fluorescence lifetime decays.

Secondly, let us summarize the biologically relevant findings we made employing this new method:

– the mean calcium concentrations in cells of the B lineage differ dependent on the stage of affinity maturation they are in

– the interplay of different signaling cues leads to fluctuations of cytosolic Ca^2+^ concentrations that span several hundred nM

– a B cell subset with high cytosolic calcium is arising during GC reaction

– a subset with even higher cytosolic calcium is present among plasma blasts

Based on these findings, we propose the exciting possibility, that selection of high affinity B cell clones in the GC is coupled to distinct concentrations of Ca^2+^ within the cells, controlled and kept within thresholds by co-stimulation and metabolic turnover. Additionally, to unravel the interplay between signaling and other cues for calcium concentration changes, like intracellular release from ER or mitochondria, is a challenge, which we would like to meet with our research in the future. It is known that there is a link between Ca^2+^ as a mediator of cell-intrinsic and –extrinsic stressors, guiding vital processes like autophagy or the unfolded protein response in order to cope with environmental changes, which may be important especially in long-lived plasma cells. Taken together, we are convinced that this system will crucially contribute to our understanding of B cell activation and differentiation in vivo.

[Editors' note: further revisions were suggested prior to acceptance, as described below.]

The reviewers, two of whom are new due to the unavailability of one of the original reviewers, appreciate your efforts to address the original concerns. The consensus is that you have succeeded in calibrating the lifetime measurements, with one caveat that the use of the lysate may not fully recapitulate the conditions in a live cell, but since it's difficult to clamp Calcium in a live cell your solution is acceptable. The experiments addressing the role of antigen in germinal centre B cell activation are considered helpful as a sensitivity test, but there is a consensus that your new text over-interprets the data presented. You have established BCR triggered Calcium flux in vitro (original data) and in vivo (new data with antigen injection), but this capability doesn't appear to help understand the heterogeneity of B cell's Calcium levels in the germinal centre or medullary cords. The reviewers can see a positive path and are happy to see another revision if you can respond to the new concerns, which largely relate to refining presentation of the new information introduced in your first revision.The calibration of the lifetime data is appreciated, but some discussion should be given to limitations of the approach, which are unavoidable, but requires caution.

We thank the reviewers and the editor for pointing out we didn’t sufficiently comment on the possible pitfalls of a fluorescence lifetime calibration of the TN-XXL construct in cell lysates, as compared to cells, in the previous version of our manuscript. In the revised version of the manuscript, we added information on the parameters which influence the fluorescence lifetime of CFP and which may differ between lysate and cytosol of a living cell. We would like to emphasize that the fluorescence lifetime of fluorophores depends on the refractive index (Strickler and Berg, *1962*, J. Chem. Phys. 37:814) and may be influenced by various other factors such as pH, temperature, or ionic strength. We have paid a lot of attention to ensure that all measurements of cell lysates were performed under conditions similar to those in the cytoplasm, except for the fact that they have been performed at room temperature instead of 37°C. While Laine et al., PLoS ONE, 2012 reported a slight decrease of Cerulean fluorescence lifetime in the TN L15 construct in the range 20°C – 50°C, we did not observe such a trend as we measured the same CFP fluorescence lifetime both at 37°C and at room temperature.

The NP-BSA induced Calcium elevation in vivo extends results with anti-Ig from Figure 2 and supports ability to detect BCR dependent Calcium ion elevation in vivo. NP-BSA can trigger apoptosis (Nossal, PMID 7753199) and that interactions with myeloid cells in the MC seems to limit antibody secreting cells numbers (Fooksman, PMID 24376270) such that signals leading to apoptosis may also be contributing to the Calcium signals observed in the system.

We are pleased that the reviewer agrees that the added experiments support the detection of BCR dependent calcium elevation. We thank the reviewer for his important comment on the relation of elevated calcium and apoptosis. We would like to point out that it is not our intention to exclude the possibility of apoptosis-induction (after addition of soluble antigen) contributing to an increased calcium concentration in the cells. This mechanism is not in conflict with what causes it, namely stimulation of the BCR, but may rather contribute to selection within the GC. In fact, it supports the assumption, that high calcium concentrations after potent BCR stimulation have to be contained via costimulatory signals, as outlined and backed up with additional references within the discussion. As the reviewer pointed out, plasma blasts in the MC may also undergo apoptosis, albeit this may be triggered by different mechanisms than in GCs. In fact, preliminary data we generated using the system in a follow-up project already point to a connection between calcium concentrations, metabolic stress and autophagy in long lived plasma cells.

This could be incorporated into the discussion of different explanations for the high Calcium subsets. It would also make sense to cite early work from Cahalan that Calcium elevation can lead to arrest of cell motility (Negulescu et al., PMID: 8630728).

The authors agree. We inserted a paragraph dealing with possible sources as well as outcomes of high calcium in cells. We thank you for introducing the relevant reference we missed, and we would like to draw your attention to figure 3 —figure supplement 1, which shows a correlation between motility and calcium elevation obtained from a single cell tracked in vivo. In our hands, cell arrest seems to appear very shortly before the rise of the calcium concentration. Since calcium, as we have added in the discussion, also affects the cytoskeleton, it contributes with some certainty to cell arrest.

But it's important to point out that demonstrating that you can detect effects of BCR engagement by injecting polyvalent antigen is not the same as showing that Calcium elevation observed in germinal centre and medullary cords in the system set up by earlier immunisation is due to antigen recognition.

Thank you. We clarified in the discussion that a direct demonstration of BCR engagement does not reflect the physiologic situation of B cells taking up AG from immune complexes bound to the FDC surface.

In fact, the apparent lack of effect of the BTK inhibitor doesn't support a role for BCR signalling in the heterogeneity of Calcium.The new data in Figure 5a is appreciated as it attempts to address the role of BCR signalling using a BTK inhibitor and injecting NP-BSA as a BCR ligand. The description of the experiments suggests that you have performed a very demanding time-course with a baseline, BTK inhibitor injection and then NP-BSA injection. Could this data be shown- similar to Figure 4g, but with breaks where the injections were performed.

Thank you for this idea and giving us the chance to present additional data (see Figure 5 —figure supplement 3): the time-dependent plots in fact are presenting two or three subsequent measurements of the same area, that is, the same GC, with breaks of about 5 min when injections were performed, indicated by interruption of the time axis. We have to add that, in our set up, it is not possible to inject during an ongoing measurement, so the cells tracked in each segment of the graph are to be treated as separate objects. We tried to be clear about this by adding text in the legend and methods, as well as to indicate the interruptions by grey dashed lines. Nevertheless, we are confident that an increase in baseline calcium levels is visible after intravenous application of NP-BSA. A slight decrease of calcium levels after ibrutinib injection, as well as a missing elevation of the calcium concentration after subsequent NP-BSA injection can also be recognized, which supports BCR stimulation with AG as one possible calcium mobilizing pathway in GC B cells.

There are questions about how long it takes these agents to get to the lymph nodes after IV injection. The inhibitor may be fine, but I'm not sure the NP-BSA will get to the germinal centre in a lymph node by this route as it will need to leak through the vasculature into the peripheral tissues and then drain to the LN. Do the authors know of data on BSA pharmacodynamics that would suggest it gets into the germinal centre in minutes after IV injection? I know of data for peptides getting into the T cell zone rapidly, but not for an intact protein that is maintained in the blood by FcRn. I would think that marginal zone of spleen would be the main point of entry for IV protein into secondary lymphoid tissues.

We appreciate this question, which brings up an important point regarding the reliability of our data. We added text citing work from Roozendaal et al., where transport of antigen into B cell follicles is characterized based on size exclusion. The study shows that small antigen is transported rapidly to the lymph node B cell follicles via conduits. The authors state that they can detect fluorescently labeled AG of less than 70kDa in the follicles within less than 2 minutes after subcutaneous injection. Since BSA has a molecular weight of about 66kDa, we can assume to achieve AG-mediated BCR simulation a few minutes after injection. For intravenous application, delivery to the LN is expected to be even faster.

As mentioned above, you need to think carefully about reasonable interpretations of these data and revise the discussion regarding possible sources of Calcium heterogeneity.

We agree and have accordingly revised the discussion. The felt bias towards BCR signaling was not our intention, instead, we tried to emphasize that calcium heterogeneity or mobilization is not to be equaled with signaling per se; and that our read-out is indirect. However, we think our data demonstrate a contribution of BCR-activated calcium flux to overall heterogeneity in cytoplasmic calcium of B lineage cells, and can serve as proof-of-principle for our system. We have substantially revised the discussion and we hope that this has contributed to make clear that we assume that intracellular calcium in B cells is influenced by many factors, some of which we discuss in more detail.

[Editors' note: further revisions were suggested prior to acceptance, as described below.]

Your last revision didn't appear to address the reviewers concerns, but the Reviewing Editor asked for another opinion in an attempt to pinpoint and articulate the key issues related to calibration.

We thank the editors and the reviewers’ panel for giving us the opportunity to clarify the open issues regarding the validity of our calibration using phasor-based FRET-FLIM of the calcium-sensor TN-XXL in the cytosol of B lymphocytes, for intravital use. Especially, the comments of the new reviewer helped us to understand the issues the reviewers had with the description of our calibration method. In this way, we had the chance to improve our manuscript with the support of an expert reviewer in phasor analysis and FLIM, underlining the interdisciplinary character of the study. Following the suggestions of the new reviewer, we provide in the new version of the manuscript a thorough description of the new calibration algorithm, which resembles the features of the algorithm proposed by Celli et al., Biophys. J, 2010. Also, we provide more information on the FRET-FLIM method itself, provide pixel-based phasor plots of FRET-FLIM data acquired extracellularly, in vitro and in vivo, and elaborate more on the advantages and shortcomings of the method for intravital calcium imaging in the cytosol of B lymphocytes, in YellowCaB mice.

The opinion below agrees that the revision didn't address reviewer concerns, but suggests a different approach to the Phasor calibration that may help resolve this issue, although it would still require some adaptation from the original use with a Calcium indicator to the current use with a Calcium biosensor. The Reviewing Editor hopes that this will be helpful in focusing a final revision that can be resent to all active reviewers, but please feel free to send a revision plan if you want feedback before undertaking major work."I found this article quite confusing in regard to the calcium calibration and the proper citation of the phasor method. The authors fail to cite the paper using the phasor approach for the calcium calibration "Celli A, Sánchez SA, Behne MJ, Hazlett TL, Gratton E, Mauro TM. The epidermal Ca(2+) gradient: measurement using the phasor representation of fluorescent lifetime imaging. Biophys J. 2010; 98(5): 911-921. PMCID: PMC2830439. This paper addresses the issue of the calibration of indicators, not biosensors. It uses a different method based on the position of the data on the phasor plot rather than attempting a calibration that cannot be done due to the diversity of the calibration solutions with respect to the environment where the biosensor is located, which is a major point of the previous comments. The only phasor plots in this paper are in figure 4, where the phasor of the biosensor is located but in my opinion the calibration method suggested in the paper by Celli et al. is not applied. This is important for the overall interpretation of the results and the conclusion of this article."

We agree with the new and the previous reviewers that by presenting the calibration algorithm only in time-domain, may have been not sufficient to highlight the validity of our calibration for its use in living B cells and in lymph node tissue, at various tissue depths. In the revised version of the manuscript, we included a section dedicated to the calibration of the TN-XXL FRET-based Ca-sensor in B lymphocytes for its use to monitor cytosolic calcium levels in lymph nodes of YellowCaB mice in vivo.

In this section, we emphasized that the TN-XXL biosensor is exclusively expressed in the cytosol of B lymphocytes and not in other, more heterogeneous compartments such as high-Ca^2+^ organelles (ER, Golgi apparatus) or the extracellular space. This information is particularly important since, as already known from the pioneering work of Gregorio Weber (Jameson et al., 1984; Gordon, Jameson, 1969), fluorescence decays of fluorophores – and by that both fluorescence lifetimes in time-domain and phase vectors in frequency-domain – may change due to alterations in physical and chemical properties of the environment surrounding the fluorophore molecules. In the revised version of the manuscript, we now emphasize how refractive index, ion strength, pH value, temperature, oxygen levels may impact on the fluorescence decay of CFP in the TN-XXL construct in B lymphocyte lysates, as compared to the cytosol of these cells. We also discuss the fact that intravital imaging of lymph node tissue at different depths may also lead to changes in the CFP fluorescence decay (e.g. due to wave-front distortions, scattering or tissue autofluorescence).

In order to assess the effect of such experimental artifacts on our results, we established, next to the previously proposed time-domain formalism to calculate cytosolic calcium levels in B lymphocytes expressing TN-XXL from FRET-FLIM data (Equation (3,4)), a formalism based only on the phasor approach of the same data. This approach is similar to that published by Celli et al., 2010 for the Calcium Green dye CG5N, but adapted to FRET-FLIM of the TN-XXL construct as follows:

– Similar to the work of Celli et al., also in our case, we detect only the free calcium with our biosensor, so that the chemical equilibrium is: Ca^2+^ + TN-XXL↔ Ca^2+^TN-XXL and its dissociation (equilibrium) constant *K_d_* can be written as [Ca^2+^TN-XXL]/[Ca^2+^][TN-XXL]. (Equation (1,2) in the manuscript). The *K_d_* value was determined, as also proposed by Celli et al., alone from the FRET-FLIM data in lysates of defined Ca^2+^ concentrations. Since cytosol and lysate of B lymphocytes share a similar composition, we also consider this approach to be sufficiently accurate, as also stated by Celli et al.

– We replaced in all following equations the unbound dye state ([CG5N]) by the unquenched state of the donor CFP ([TN-XXL]) and the fully bound dye (saturated with calcium) with the fully FRET-quenched state of CFP ([Ca^2+^TN-XXL]). In this way, we deduced the relation between free calcium concentration and the measured phase vector in each pixel of the image (Equation (5)), which depends on the phase vectors and brightness values of CFP in the two extreme TN-XXL states, i.e. no bound calcium and fully saturated by calcium.

– We calculated the brightness values of the two extreme states of CFP, i.e. TN-XXL and Ca^2+^TN-XXL, as the product of their fluorescence lifetimes, fluorescence decay rate of CFP in vacuum *k_F_*, (together fluorescence quantum yield) and active two-photon absorption cross-section of CFP under excitation at 850 nm, *_CFP_*. Since neither the excitation (given by *δ_CFP_*) nor the fluorescence rate *k_F_* in vacuum are influenced by the molecular surrounding of a fluorophore molecule, these values are the same for all states of CFP. In the Equation (6,7) only the fluorescence lifetime reflects the impact of the molecular environment on the brightness value. Since the exact fluorescence lifetimes of CFP in the extreme TN-XXL states (2312±54 ps and 744±90 ps, respectively), as measured in cell lysate at known free calcium concentrations, could also be measured under our in vivo conditions, in B lymphocytes in lymph nodes of YellowCaB mice, we conclude that the ratio of the brightness values for the extreme TN XXL states remains the same for lysate, B lymphocytes in culture and in live lymph node tissue.

– We show phasor plots of CFP fluorescence decays in an image:

i. In lysate solutions of YellowCaB B lymphocytes at defined free calcium concentrations (0 nM, 39 µM and 602 nM) – Figure 4c,

ii. In YellowCaB B lymphocytes in three different lymph nodes (three different animals), acquired in vivo, Figure 4e, and

iii. In YellowCaB B lymphocytes in cell culture, (Figure 4 —figure supplement 1).

In all phasor plots, we represented the extreme states measured in lysate as blue cloud (0 nM free calcium) and as red cloud (39 µM free calcium). In all three experimental setups (extracellular, cell culture and in vivo) the orientation of the calibration segment connecting the blue cloud of “no calcium” (TN-XXL) and red cloud of “full calcium saturation” (Ca^2+^TN-XXL) in the phasor plot remains the same. Under in vivo conditions we could not induce a complete shift of the phasor cloud, i.e. in all B lymphocytes of the popliteal lymph node, towards intracellular calcium saturation, due to animal welfare reasons. However, we repeatedly detected that parts of the phasor clouds in lymph nodes reach out to the position of calcium-saturated TN-XXL (red cloud), indicating that both orientation and length of the calibration segment determined in lysates remains valid under in vivo conditions.

– By showing the phasor plots of CFP fluorescence in B lymphocytes (Figure 4f), in different depth layers of a lymph node, encompassing B cell follicles and medullary cords (representative data of n = 3 mice), we could confirm that the data comply with the requirements imposed by the calibration, independent of tissue depth.

– We further acquired phasor plots of endogenous signal in lymph nodes of non-fluorescent C57Bl/6J mice (Figure 4g), to assess the impact of this signal on our cytosolic calcium results. Generally, the endogenous signal after excitation at 850 nm in B cell follicles and medullary cords areas was low. Its phasor cloud was located around the origin (0;0) of the phasor plot, indicating that mainly electronic noise of the device is responsible for this signal rather than optical signal originating from the sample, e.g. autofluorescence. Hence, if the excitation power is not sufficient to induce an appropriate fluorescence signal of CFP in B lymphocytes expressing TN-XXL, the phasor cloud of CFP fluorescence will be a linear combination of three reference positions: the two extreme positions of CFP fluorescence (from TN-XXL) and the origin of the phasor plot. This would result into a shift of the phasor cloud away from the calibration segment towards the origin (0;0). As shown in Figure 4c, 4e and in Figure 4 —figure supplement 1, this is not the case in our experiments.

Concluding, both Equation (5) for the phase vector in each pixel and Equation (11) for calculating free calcium concentrations, i.e. the key equations of the phasor-based formalism, are valid for extracellular (lysates) as well as cellular measurements, in vitro and in vivo, in our case.

Additionally, we now stated in the manuscript the dynamic range of the biosensor, giving the absolute Calcium concentrations that we could measure based on our FRET-FLIM data, i.e. in the range between 100 nM and 4 µM free calcium.

We further compared both formalisms (time-domain and phase-domain) in order to test whether they are equivalent. The discrepancies between the calcium level values calculated with these two formalisms (Equation 4 vs. Equation 11) were lower than 5 % in all cases, presumably due to numerical uncertainty caused by logarithmic calculation in Equation (4).